# Functional Safety for Language Models

## Abstract

Large language models (LLMs) are increasingly used for document and code editing, yet standard flat editing workflows generally do not expose formal assurances about the scope, structure, or side effects of their modifications. We introduce *Functional Safety*, a hierarchy-aware editing architecture that formalizes LLM-driven edits as typed plans over explicit hierarchies with deterministic execution. A stochastic planning stage operates on an explicit hierarchical representation and emits a structured plan of typed operations that separate structural reorganization from bounded content generation. We analyze each step with two footprints: a *structural footprint* (nodes whose relations may change) and a *payload footprint* (nodes whose local content may change), and the guarantees are scoped per step. Execution is performed by a deterministic, structure-constrained component that enforces locality, guards protected regions, preserves byte-for-byte payload outside each step's payload footprint, and confines structural changes to each step's structural footprint under the stated assumptions. We formalize the architecture, specify its invariants, and prove a Deterministic Safety theorem for correctly extracted hierarchies and valid symbolic plans. Empirical evaluations on long-form document rewriting, code refactoring, and multi-page policy briefs show improved strict success rates and lower observed side-effect rates relative to representative ReAct-style tool agents in the evaluated conditions. These results suggest that principles from functional programming—explicit structure, composability, and controlled side effects—provide a conditional and auditable foundation for LLM-driven editing, while leaving extraction, planning, and worker conformance as explicit reliability conditions.

## 1  Introduction

Modern large language models (LLMs) can draft, revise, and reason over text and code across many settings, yet their behavior on structured editing tasks remains fragile. Human edit intent is typically localized and specific, while the remainder of the artifact is expected to remain invariant. By contrast, many LLM editing workflows operate over flat token sequences where structure is only implicit, so locality is inferred rather than enforced and formal scope guarantees are generally not exposed. Even when prompted to make narrowly scoped changes, such systems often regenerate long spans with no explicit notion of hierarchy or change semantics. As a result, even minimal edits can induce structural drift and side effects outside the intended scope.

These failures are more than minor inconveniences. In settings such as legal drafting, scientific writing, and code maintenance, users often need strong assurances that an edit request will not silently alter definitions, assumptions, or program behavior outside the requested region. Two factors are entangled in current systems: open-ended generative decoding, which can re-sample content even for localized requests, and flat token-level representations, which provide no explicit boundaries for where changes are permitted. Together they make it difficult to bound edits or reason about their impact.

From an autoregressive model's perspective, a paragraph, a function, or a theorem is merely a contiguous sequence of tokens. The model typically infers boundaries and structure implicitly, and therefore does not explicitly enforce invariants such as preservation of protected regions, structural well-formedness, or strict edit locality. In domains where precision is essential, such implicit reasoning may be too limited. The

central problem is the absence of a principled mechanism for enforcing edit locality and explicit structural constraints during transformation.

This paper describes *Functional Safety*, a hierarchy-aware architecture intended to provide stronger behavioral properties for LLM-driven editing. At a high level, the approach factors editing into two primary phases within a multi-stage pipeline. A stochastic planning stage works over an explicit hierarchical representation of the document or codebase, selecting which units to modify and how. A separate, functionally pure executor then applies these plans using deterministic tree transformations that respect invariants about protected regions, structure, and layout. By localizing randomness to the planning stage and treating execution as a pure function, the architecture makes it possible to reason about side effects in a way that is largely model-agnostic. We make this precise by distinguishing a per-step *payload footprint* (local content that may change) from a per-step *structural footprint* (nodes whose relations may change); plan-level unions summarize aggregate impact.

More broadly, the framework integrates insights from functional programming and hierarchical models of human cognition. Functional languages emphasize purity, composability, and explicit control of side effects; hierarchical cognitive theories emphasize chunking, multi-level representation, and compositional operations over symbols. Functional Safety combines these ideas to obtain a simple but expressive abstraction for LLM-mediated edits: plans are stochastic, but their interpretation is deterministic and structure-constrained. Although errors cannot be eliminated, they can be confined to explicit interfaces and bounded in scope when extraction, planning, and worker-output contracts hold.

## 1.1 Contributions

This paper makes four contributions toward a principled notion of functional safety in language models:

- **Functional Safety procedure.** We introduce a hierarchy-aware editing procedure that separates *planning* from *execution*. Documents and programs are represented as graphs of semantic units (sections, paragraphs, methods, blocks). A stochastic planning stage operates over this hierarchy to propose local edits, while a deterministic executor applies those edits using pure tree transformations. Stochasticity enters the system through two channels: the planning stage, which selects semantic units and proposes edits, and worker modules, which may call language models to generate content within plan-selected regions. In this formulation, accepted worker outputs are applied only through executor-scoped typed operations, while planner-selected structural changes are checked against each step's structural footprint when extraction, planning, and worker-output contracts hold. A deterministic executor applies structural checks and merges results, confining model variability to each step's payload footprint while constraining structural changes to each step's structural footprint (with plan-level unions summarizing aggregate impact).

- **Verified semantics.** We provide a formal model for the hierarchical representation, the planning function, and the executor. Within this model we state and prove a deterministic executor theorem: if the input hierarchy is correct, the planning stage respects a small set of interface contracts, and worker outputs satisfy the plan-determined local output contract, the executor applies accepted replacements only through the typed operation and preserves content outside each step's payload footprint in the formal model. This yields a notion of functional safety that is independent of any particular language model but conditional on the stated interface contracts.

- **Empirical coverage across domains.** We instantiate the architecture in a practical system and evaluate it on large-document editing and non-trivial code refactoring tasks, comparing against a representative ReAct-style tool agent with context-matching patches via an observe–think–act loop. Functional Safety improves observed strict success rates and lowers observed side-effect rates relative to the baselines in the evaluated single-document conditions, with the largest gains on structural and multi-location edits. The comparison indicates that hierarchy-aware planning and deterministic execution can provide reliability benefits that patch-based editing without explicit structure does not consistently match on these tasks.

- **Bridge to emerging literature.** Finally, we relate Functional Safety to work on long-context transformers, hierarchical cognition, tool-augmented language models, and functional programming. We argue that separating stochastic planning from deterministic execution provides a common language for these lines of research and outline how the proposed framework can serve as a foundation for a broader research program on side-effect-aware LLM systems.

## 1.2 Background

The limitations of token-level reasoning for structural editing have been widely recognized. Standard approaches treat documents as flat sequences of subword tokens and apply autoregressive generation to entire spans, even when the requested change is local. Without an explicit representation of sections, paragraphs, code blocks, or method boundaries, the model has no way to distinguish between regions that may be altered and regions intended to remain invariant. This mismatch between sequence-level operations and structured documents accounts for the unintended side effects observed in practice.

Human cognition, by contrast, relies heavily on hierarchical abstraction when manipulating complex artifacts such as proofs, programs, or manuscripts. People reason in terms of chapters, sections, lemmas, functions, and blocks; they perform local edits at one level of the hierarchy while keeping higher-level structure fixed. This style of reasoning supports both global coherence and local flexibility, and it suggests that reliable machine editing may call for explicit representations of semantic units rather than raw token streams.

Several strands of machine learning research move in this direction. Work on long-context models and efficient attention mechanisms extends the span of text that can be processed in a single forward pass, but typically leaves the underlying representation flat and token-based (Beltagy et al., 2020; Zaheer et al., 2020; He et al., 2024). ReAct-style tool agents and software-engineering agents provide a representative pattern for agentic editing: an LLM observes artifact state, reasons about what to change, and applies patches or span edits through tool calls (Yao et al., 2023; Yang et al., 2024; Wang et al., 2025b). This approach improves on flat generation for simple localized edits in our experiments, but still operates over linear text without explicit hierarchy, and its reliability degrades on the evaluated structural operations and multi-location edits. Other work explores constrained decoding and programmatic scaffolding to shape model outputs (Willard & Louf, 2023; Dong et al., 2025b), while layout-aware document-understanding systems infer useful document structure from visually rich inputs (Wang et al., 2024); these approaches are complementary but do not by themselves provide a formally specified execution layer that isolates and controls edit side effects. Existing code-refactoring tools, meanwhile, provide strong assurances but are tailored to specific languages and do not leverage LLMs for high-level planning.

The approach developed in this paper builds on a different combination of ideas. We assume that language models are powerful but noisy planners over latent structure, and we place a deterministic, functionally specified executor between their plans and the underlying document. By making the hierarchical representation explicit and the executor pure, we obtain a system in which structural invariants can be stated and proven once, then reused across models and domains. Functional Safety thus aims to improve not only performance but also the correctness, predictability, and safety of LLM-driven editing.

## 2 Formal Model

In this section we present a mathematical framework for the hierarchy-aware editing architecture. Our goal is to formalize the core components—hierarchical representations, the router and planner, stochastic worker modules, and the deterministic executor—in a way that makes their roles, constraints, and interactions explicit. This formalization isolates the sources of stochasticity, specifies the symbolic structures on which plans operate, and establishes the invariants used for reliable, content-side-effect-bounded transformations. This is one formalization of the approach; alternative formulations or operator sets are also possible.

**System Components.** The architecture consists of five distinct components:

- **Extractor** $C$: a deterministic or validated host procedure that constructs the hierarchy from the raw input. The formal guarantee below is conditional on this hierarchy being lossless and aligned with the document units relevant to the edit.
- **Router** $g$: a lightweight model that maps a user request and node summaries to either (i) a typed plan $P$ in the edit algebra $\mathcal{O}$ or (ii) a delegate directive for a permutation-only symbolic planner. In this formalization, the router does not itself perform structural edits.
- **Planner** $f_\theta$: a stochastic LLM-based symbolic planner invoked when the router delegates a pure permutation; it emits a section ordering (or equivalent plan fragment) that the host expands into a MOVE.
- **Workers** $\{W_j\}$: optional stochastic LLM modules that perform bounded local transformations on extracted spans. In this formalization, a worker is not given global structure and its scope is restricted to the node selected by the plan.
- **Executor** $E$: a deterministic executor for symbolic plans and accepted local replacements. The plan and current hierarchy determine any worker requests (target node/span, instruction, optional crop, read-only context, and output contract). Workers may produce stochastic candidate text, but the theorem-level executor consumes only accepted replacement outputs and applies them through the typed plan while enforcing structural invariants and payload preservation outside each step's payload footprint.

Stochasticity may arise in the router, the delegated planner (when used), and the workers. Conditional on a validated plan and the accepted local replacements produced for the plan-determined worker requests, the executor itself is pure and deterministic.

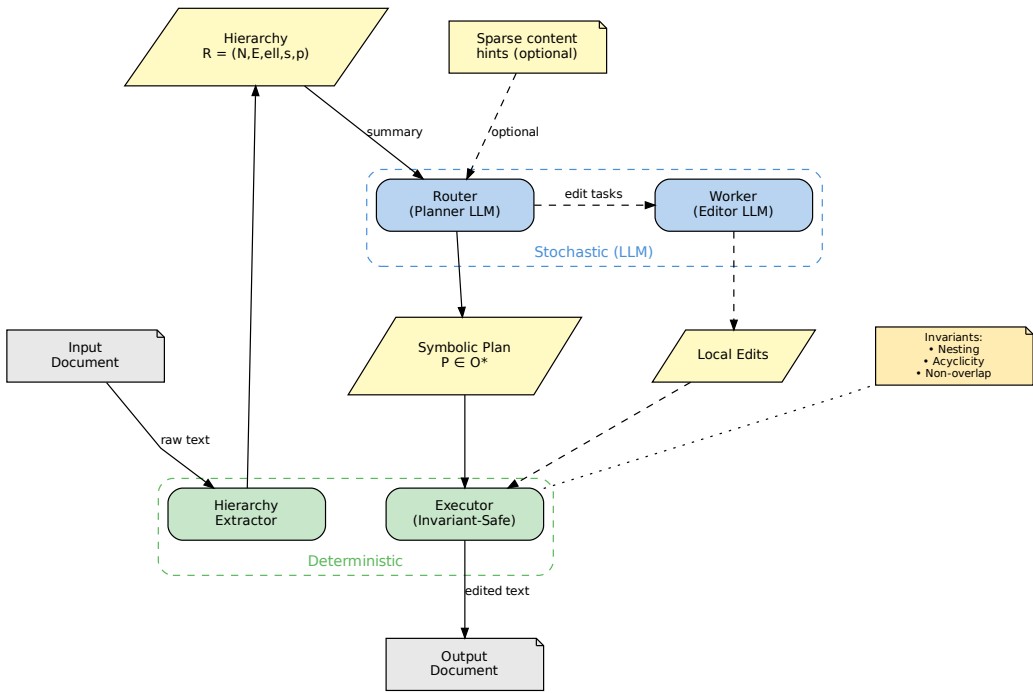

Figure 1: System architecture separating stochastic LLM components (Router/Planner, Workers) from deterministic components (Extractor, Executor). The symbolic plan $P$ serves as the interface between the two domains, enabling formal reasoning about execution behavior.

### 2.1 Raw Input and Rendered Positions

Let the raw input document or code corpus be represented as an ordered sequence of rendered units

$$X = (x_0, x_1, \ldots, x_{T-1}).$$

The unit can be bytes, characters, or tokens as chosen by the host implementation; the formal model only requires a deterministic total order of rendered positions. Beyond initial parsing, the architecture does not operate directly on $X$. For the supported deterministic extractors used in the structured experiments, extraction is lossless: the hierarchy $R$ satisfies $\text{render}(R) = X$. When we write $C(X)$ below, $C$ denotes this byte-preserving representation canonicalization: it assigns payload ownership, spans, and optional gap nodes, but it does not normalize the input bytes during initial extraction.

### 2.2 Hierarchical Representation

A hierarchical representation $R$ can be described both (i) compositionally as a tuple and (ii) declaratively through its constituent components.

**Tuple Form.**
$$R = (N, E, \ell, s, p),$$

**Component Form.**

- $N$: a finite set of nodes corresponding to semantic units (paragraphs, sections, functions, etc.).

- $E \subseteq N \times N$: directed edges defining parent–child relations.

- $\ell : N \to \mathcal{L}$: labeling function assigning each node a semantic type.

- $s : N \to \{(i,j) \mid 0 \le i \le j \le T\}$: span function mapping each node to a half-open interval $[i,j)$ in the rendered document.

- $p : N \to X^*$: payload function storing the *local* content of each node (text owned by the node outside its child spans, e.g., a paragraph's text, heading tokens, or structural separators).

**Paragraph boundary convention.** For prose formats, paragraph boundaries are defined by blank lines. Initial extraction preserves the input bytes and represents existing inter-paragraph whitespace as payload, either inside paragraph spans or as explicit gap nodes. If a later structural edit makes two paragraph nodes adjacent without a blank-line separator, an execution-time separator policy may insert the minimal separator as an in-footprint payload change. A strict-raw mode disables this repair when byte identity is required, at the cost of possible paragraph merges after moves. Appendix D describes these policies and their footprint interaction.

**Rendered Content.** We define a deterministic render function

$$\text{render}_R(n) = p(n) \parallel \text{render}_R(c_1) \parallel \cdots \parallel \text{render}_R(c_k),$$

where $(c_1, \ldots, c_k)$ are the ordered children of $n$ and $\parallel$ denotes concatenation (including any format-specific separators encoded in payload or node metadata). The document text is $\text{render}(R) = \text{render}_R(\text{root})$. Spans are derived by rendering: $s(n)$ is the interval occupied by $\text{render}_R(n)$ inside $\text{render}(R)$. Rendered content can change when children move even if a node's local payload does not, which is why non-interference is stated in terms of $p(n)$ below. The displayed equation is schematic: implementations with inter-child or trailing local payload represent those segments as explicit gap nodes or prefix/trailing payload as described in Appendix D. The theorem only requires deterministic rendering and explicit payload ownership.

**Well-Formedness Constraints.**

- **Nesting**: Parent spans contain child spans.

- **Acyclicity**: $(N, E)$ contains no directed cycles.

- **Non-overlap**: Nodes not in an ancestor–descendant relation have disjoint spans.

**Identity and Reconciliation.** Each node $n \in N$ carries a stable entity identifier $e(n)$ that persists across edits and moves, and a content identifier derived from its local payload. When a refresh re-extracts a hierarchy from text, the executor computes a reconciliation map $\chi$ from prior nodes to refreshed nodes. The reconciliation is *structure-first*: parent/position anchors and spans are primary signals, and payload similarity is used only as a tie-breaker. Outside the allowed payload footprint, $\chi$ must be total and unambiguous; if any node outside the footprint cannot be matched uniquely, execution fails (fail-closed). Inside the footprint, unmatched nodes are permitted and receive fresh entity identifiers. Merge and split events create new entity identifiers and record lineage edges so continuity is preserved without reusing prior identities. The formal model assumes strict payload identity outside the payload footprint on the byte-preserving hierarchy representation. Optional executor policies that allow whitespace-normalized matching for soft adjacency nodes, or that insert missing paragraph separators after structural edits, are implementation variants and are not part of the core theorem unless explicitly specified.

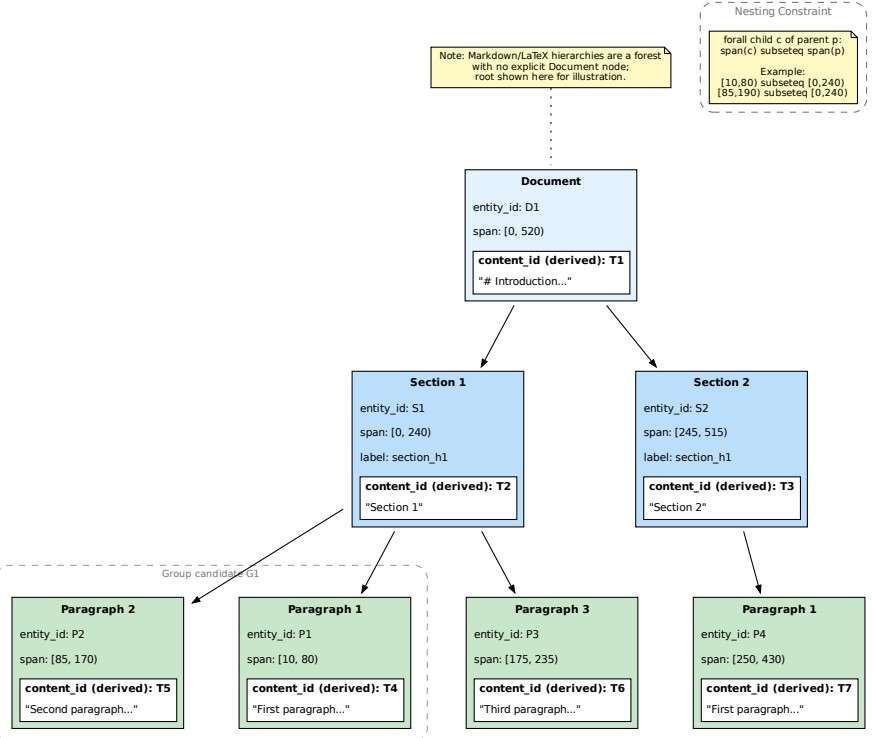

Figure 2: Example hierarchical representation $R$ showing a document with two sections and three paragraphs. Each node stores a 0-indexed half-open rendered-position span. The nesting constraint assumes child spans are contained within parent spans (e.g., $[10, 80) \subseteq [0, 240)$).

### 2.3 Edit Algebra Formalization

Let $S = (s_1, \ldots, s_m)$ be the top-level nodes (e.g., sections). The edit algebra is:

$$\mathcal{O} = \{\text{MOVE}, \text{EDIT}, \text{INSERT}, \text{DELETE}, \text{ANNOTATE}\}.$$

We use MOVE for deterministic section permutations ($\text{MOVE}(\pi)$) and subtree relocation ($\text{MOVE}(n, \text{pos})$); "reorder" tasks are instances of $\text{MOVE}(\pi)$.

**Operator Semantics.**

- $\text{MOVE}(\pi)$: $\pi$ is a permutation of $(s_1, \ldots, s_m)$. The executor deterministically permutes section buffers; no worker is invoked.

- $\text{MOVE}(n, \text{pos})$: relocate the subtree rooted at node $n$ to position pos (parent/index or anchor before/after). The executor performs a deterministic structural move and preserves payload content byte-for-byte; paragraph separators are handled by the separator policy.

- $\text{EDIT}(s_i, r, \theta)$: $s_i$ is a node identifier, $r$ is a natural-language instruction, and $\theta$ is an optional crop $(\alpha, \beta)$. If $\theta$ is provided, the worker receives the cropped substring of $\text{render}_R(s_i)$ and $r$; the executor splices the result back into $s_i$. If the plan supplies replacement content directly, the worker can be bypassed and the executor splices the provided content. If $\theta$ is empty, the worker (or the supplied content) replaces the rendered content of that node; if $s_i$ has children, the executor re-extracts the subtree rooted at $s_i$ after the replacement to rebuild its internal structure and payloads.

- $\text{INSERT}(\text{content}, \ell, \text{pos})$: creates a new node with payload content and label $\ell \in \mathcal{L}$ at position pos. The position specifier pos can indicate: (i) a parent node and child index, (ii) an anchor sibling with before/after placement, or (iii) a root-level index. The new node is assigned a fresh identifier $n_{\text{new}} \notin N$.

- $\text{DELETE}(n)$: removes node $n \in N$ and all of its descendants from the hierarchy. Parent–child links are updated to excise the deleted subtree.

- $\text{ANNOTATE}(n, \ell)$: update node $n$'s label/metadata to $\ell$ without changing its payload (e.g., relabel a heading as a sub-heading in the hierarchy). If rendering in a format depends on the label token itself, this operation is paired with a corresponding EDIT.

**Insert and Delete.** The INSERT and DELETE operators are structural primitives that modify the node set $N$. INSERT adds a new node at a specified position within the hierarchy—either as a child of an existing node, as a sibling relative to an anchor node, or at the root level. The content of the inserted node may be provided directly by the plan or authored by a worker LLM when the plan specifies a reason and read-only context; in either case, the structural placement is deterministic. DELETE removes a node and its entire subtree, updating parent references to maintain well-formedness. Together, these operators enable the planner to express additive and subtractive structural edits, complementing the content-modifying EDIT and the purely structural MOVE.

### 2.4 Plan Validity and Typing Rules

A plan is a finite sequence $P = (o_1, \ldots, o_k)$ with $o_i \in \mathcal{O}$. The special directive DELEGATEPERMUTATION is also permitted. Here $\mathcal{O} = \{\text{MOVE}, \text{EDIT}, \text{INSERT}, \text{DELETE}, \text{ANNOTATE}\}$. Let $\mathcal{P}$ denote the set of well-typed symbolic plans over $\mathcal{O}$, and let $\mathcal{R}$ denote the set of well-formed hierarchies. Typing judgments take the form

$$R \vdash o_i : R \to R_i,$$

with $R_0 = R$, and extend compositionally to $R \vdash P : R \to R_k$.

A symbolic plan $P = (o_1, \ldots, o_k)$ is *well-typed* for hierarchy $R = (N, E, \ell, s, p)$ with top-level nodes $S = (s_1, \ldots, s_m)$ iff each operator $o_i$ satisfies the corresponding typing rule:

(a) MOVE($\pi$) operator is well-typed for $S = (s_1, \ldots, s_m)$ iff the multiset of elements in $\pi$ equals the multiset of elements in $S$:

$$\{\{\pi_1, \ldots, \pi_m\}\} = \{\{s_1, \ldots, s_m\}\}.$$

Equivalently, the map $i \mapsto \pi_i$ defines a bijection from $\{1, \ldots, m\}$ onto $S$.

(b) MOVE($n$, pos) is well-typed iff

- $n \in N$ is a valid node,
- pos specifies a valid position (parent exists, anchor exists, or root-level index is valid),
- if pos is parent-based, then the parent is not in the subtree of $n$,
- if pos is anchor-based, then the anchor is not in the subtree of $n$ and the parent inferred from the anchor is valid.

(c) EDIT($s_i, r, \theta$) is well-typed iff

- $s_i$ is a valid node (i.e. $s_i \in N$),
- $r$ is a natural-language instruction when worker generation is used; if the plan supplies concrete replacement content, $r$ may be empty and the worker is bypassed, and
- if a crop $\theta = (\alpha, \beta)$ is supplied, then

$$(\alpha, \beta) \subseteq s(s_i),$$

  where $s(s_i)$ denotes the rendered-position interval associated with node $s_i$.

(d) INSERT(content, $\ell$, pos) is well-typed iff

- $\ell \in \mathcal{L}$ is a valid semantic label,
- content is a (possibly empty) string, and
- pos specifies a valid position:
  - if pos = (parent_id, index), then parent_id $\in N$ and $0 \leq$ index $\leq |\text{children}(\text{parent\_id})|$,
  - if pos = (anchor_id, rel) with rel $\in \{\texttt{before}, \texttt{after}\}$, then anchor_id $\in N$,
  - if pos = ($\texttt{root}$, index), then $0 \leq$ index $\leq m$.

(e) DELETE($n$) is well-typed iff $n \in N$.

(f) ANNOTATE($n, \ell$) is well-typed iff $n \in N$ and $\ell \in \mathcal{L}$.

(g) DELEGATEPERMUTATION is well-typed iff it expands uniquely to a valid MOVE. Formally, the plan supplies a permutation $\pi$ of $S$, yielding the operator MOVE($\pi$) which satisfies rule (a).

## 2.5 Executor Semantics

The executor applies the operator semantics defined in Section 2.3 deterministically and in sequence, using accepted worker outputs only where the plan requires generation and enforcing the structural constraints described below.

In the router–hybrid pipeline, MOVE($n$, pos) is used for section and paragraph relocation; paragraph moves require paragraph anchors or a parent section, while MOVE($\pi$) remains the dedicated section permutation operator.

The executor checks nesting, acyclicity, non-overlap, and payload non-interference after each operation. When span alignment is required, it refreshes the hierarchy deterministically, and when an EDIT can change structure, it re-extracts the affected subtree.

**Unified Position Specifiers.** The executor uses a unified positioning model for both INSERT and internal move operations. A position specifier pos can take three forms:

1. *Index-based*: $(\text{parent\_id}, k)$ places the node as the $k$-th child of node parent_id.

2. *Anchor-based*: $(\text{anchor\_id}, \text{rel})$ with $\text{rel} \in \{\texttt{before}, \texttt{after}\}$ places the node immediately before or after the anchor sibling. The target parent is inferred from the anchor's parent.

3. *Root-level*: $(\texttt{root}, k)$ places the node as the $k$-th top-level node.

This unified model allows planners to express both absolute positioning ("make this the third paragraph of section 2") and relative positioning ("insert this after the introduction paragraph") using the same underlying mechanism.

## 2.6 Stochastic Planning Function

The planning stage is stochastic; we can model it as a function

$$P = f_\theta(X, R, \omega),$$

where $\omega$ captures all randomness. In this formalism, operations refer to symbolic node identifiers. In the router–hybrid implementation, the router emits a plan $P$ directly or a delegate directive; when delegation occurs, a symbolic planner provides a permutation that is expanded into a MOVE and spliced into the plan.

The planning stage is responsible for emitting a well-formed symbolic plan in the edit algebra. In practice, this entails the underlying model producing syntactically valid operations, but questions of empirical reliability are addressed in the experimental section.

For any validated plan, the host can deterministically derive the finite sequence of local worker requests

$$Q(R, P) = (q_1, \ldots, q_m),$$

where each request fixes the target node or crop, edit instruction, read-only context, and output contract. Worker modules produce candidate replacements $Y = (y_1, \ldots, y_m)$ for these requests. The executor accepts only outputs that satisfy the interface contract; invalid outputs are rejected or retried by the implementation and are not silently applied.

## 2.7 Deterministic Executor

Given a hierarchical representation $R$ and a validated symbolic edit plan $P = (o_1, \ldots, o_k)$, let $Y$ be the sequence of accepted local replacements for the plan-determined worker requests (empty when no operation requires generation). The executor applies the operators and accepted replacements in sequence to produce a new hierarchy,

$$R' = E(R, P, Y),$$

where $E$ is total and deterministic once both the plan and accepted worker outputs are fixed. Let $\mathcal{P}_{\text{val}} \subseteq \mathcal{P}$ denote plans that have passed schema, typing, footprint, and structural-precondition validation; in the formal model,

$$E : \mathcal{R} \times \mathcal{P}_{\text{val}} \times \mathcal{Y}_{\text{acc}} \to \mathcal{R}.$$

Candidate plans outside $\mathcal{P}_{\text{val}}$ are handled by the implementation-level validator, which returns an explicit error and performs no edit. Candidate worker outputs outside $\mathcal{Y}_{\text{acc}}$ are rejected before they become theorem-level inputs.

That is, once the initial hierarchy $R$, validated plan $P$, and accepted local replacements $Y$ are fixed, the executor's behavior is fully determined: there is exactly one resulting hierarchy $R'$. The plan determines which worker requests are made, but not the stochastic text returned by the worker. The deterministic safety claim is therefore conditional on accepted worker outputs: inside the selected payload footprint, different

valid worker outputs may produce different local text; outside that footprint, worker output is not allowed to alter the document's payload because the executor does not splice it there. Structural metadata (spans and ordering) may be recomputed deterministically as required by structural edits.

Thus, for fixed $R$, validated $P$, and accepted $Y$, the global structure of $R'$ and all payload outside the planner-selected regions are uniquely determined. The executor therefore functions as a pure, structure-constrained interpreter for symbolic plans and accepted local replacements, ensuring that accepted content side effects remain local, bounded, and auditable under the stated interface contracts.

## 3 Hierarchy-Aware Functional Safety

Functional Safety separates planning from execution. The host first builds an explicit hierarchy over the input, a router emits typed edit operations over that hierarchy, and a deterministic executor applies those operations while enforcing locality and structural validity. This section summarizes the operational flow corresponding to the formal model in Section 2.

### 3.1 Pass I: Host-Grounded Hierarchy Extraction

The host extracts a hierarchy before planning or worker calls. For structured formats (e.g., Markdown, LaTeX, and code), extraction is deterministic; for weakly structured inputs, hierarchy induction can be learned or heuristic. The extracted hierarchy is the authoritative representation for subsequent planning and execution: nodes carry local payloads and ordered children, and rendering the hierarchy recovers the document under lossless canonicalization.

To keep planning context compact, the host provides router-facing descriptors (IDs, titles, and short previews) rather than full node payloads. For tasks that require local structural detail (for example, edits inside a table or list), the executor may construct temporary local views and apply the edit within that scoped view before reintegrating it into the main hierarchy.

### 3.2 Pass II: Router-Based Symbolic Planning

The router receives the user instruction plus a compact hierarchy outline and emits a typed plan over five operators:

- MOVE: reorder or relocate existing nodes.

- EDIT: modify content in a selected node (optionally on a cropped span).

- INSERT: add a new node at a validated position.

- DELETE: remove a node and its descendants.

- ANNOTATE: relabel a node (for example, heading-to-subheading) without expanding write scope.

The router may also delegate pure permutations to a symbolic permutation routine. Context units are read-only: they can inform generation but do not expand the writable scope of an operation.

**Router/Planner Decomposition.** Figure 1 presents a unified view for readability. Conceptually, routing and delegated symbolic planning are distinct interfaces: routing chooses typed operations, while delegated planning solves permutation-only structure edits.

### 3.3 Pass III: Deterministic Execution

The executor validates each operation, computes step-wise footprints, and applies edits deterministically once accepted local replacement outputs are fixed. MOVE, INSERT, DELETE, and ANNOTATE are interpreted

directly by the executor. EDIT may require a worker request determined by the plan and current hierarchy, but any accepted replacement is spliced only at the targeted scope and then reconciled against the hierarchy.

After each step, the executor checks structural well-formedness and payload non-interference outside the allowed footprint. If an edit changes internal structure, the affected subtree is refreshed before subsequent steps. When remaining operations are no longer valid under the refreshed hierarchy, execution fails closed or replans from the updated state, depending on configuration.

Worker outputs are validated before application. Invalid outputs are rejected, optionally with bounded retry, and failures are surfaced as explicit execution errors rather than silently applied edits. This deterministic enforcement layer is the mechanism behind the executor non-interference claims in Section 4.

## 4    Functional Safety Properties

A central objective of the architecture is to reduce unintended changes to the input. By constraining edits to a small, symbolic edit algebra and executing plans through a deterministic executor, the system is designed to enforce structural and semantic properties that token-level generation does not typically provide.

**Planner Capability Assumption.**    The formal results in this section assume that the planning stage (router and/or delegated planner) emits a well-formed symbolic plan. The executor computes a structural footprint *per operation* (denoted $F_{\mathrm{struct}}(o_i)$) that is closed under every *structural* change a step may induce: moved subtrees, sibling-order changes (root order for roots), and any structural insert/delete effects produced by subtree refresh. If the plan supplies an intended structural footprint $T$, the executor checks $F_{\mathrm{struct}}(o_i) \subseteq T$ for every step and rejects the plan otherwise. If no $T$ is provided, the executor may set $T := \bigcup_i F_{\mathrm{struct}}(o_i)$ by default. When a plan passes these checks, the executor enforces payload preservation outside each step's payload footprint (defined below) in the formal model under these assumptions (Theorem 5). Worker modules that implement EDIT operations may be stochastic, but the executor only gives them the plan-determined local input and only accepts outputs matching that local output contract; the executor then splices accepted output at the declared target or crop and checks that payload outside the step footprint remains byte-identical. When re-extraction is required, it is scoped to edited subtrees when possible and escalates to parent or full-document scope only when required for well-formedness or lossless rendering. If the router outputs malformed JSON, an invalid operator, or a reference to a nonexistent node, the executor rejects the plan and performs no edit. Full-document rewrites are permitted by setting $T{=}N$, in which case the side-effect claim becomes vacuous but the same execution model applies. The router-hybrid implementation used in experiments enforces equivalent locality via protected-section invariants and paragraph-level footprint checks rather than an explicit $T$. These formal guarantees apply to the executor defined here and to any implementation that satisfies the same footprint-closure and refresh assumptions. Plan validation simulates each operation and checks invariants per step; during execution, invariants are enforced step-by-step. The executor may refresh the hierarchy for span alignment, but refresh scopes induced by EDIT are handled per step. For a fixed plan, this implies that mixed plans are equivalent to executing each step in isolation; when the system replans between steps (e.g., after a hierarchy refresh), the equivalence applies to each plan segment, not to the full replan loop.

### 4.1    Edit Algebra

The edit algebra specifies the finite set of symbolic operations that the planning stage may emit. These operators form the primary interface between the stochastic planning stage and the deterministic executor in this formulation, and thereby define the space of permissible transformations.

Let $\mathcal{O}$ denote the operator set defined in Section 2.3. It consists of five typed operators—MOVE, EDIT, INSERT, DELETE, and ANNOTATE—plus the special directive DELEGATEPERMUTATION, which expands deterministically to a MOVE after validation.

The executor interprets these operators symbolically in this formulation. Token-level generation occurs inside worker calls attached to EDIT, and unspecified spans remain untouched. This separation—symbolic structure at the top level, bounded generation at the leaves—is the foundation of functional safety.

**Implementation note.** Concrete plan schemas map to the abstract EDIT/MOVE/INSERT/DELETE/ ANNOTATE operators; the executor validates structural preconditions and span consistency before applying a plan.

**Whitespace handling (appendix pointer).** Detailed whitespace handling, canonicalization, and separator policies are documented in Appendix D.

## 4.2 Safety Invariants

We now formalize the correctness properties enforced by the deterministic executor. Let $R$ be a well-formed hierarchy, let $P \in \mathcal{P}$ be a symbolic plan, let $Y$ be the accepted local replacements for any plan-determined worker requests, and let

$$R' = E(R, P, Y)$$

be the result of executing the plan. For per-step reasoning, let $R_0 = R$ and $R_i = E(R_{i-1}, (o_i), y_i)$ denote the intermediate hierarchy after step $i$, where $y_i$ is empty for operations that do not use worker-generated replacement content. For a single operator $o$, let $\text{Targets}_o$ denote the nodes it directly references, and $\text{Targets}_o(\text{EDIT})$ the subset referenced by EDIT. Unless an edit is constrained to a strict subset of a leaf node, it may change internal structure within its scope; therefore the executor re-extracts the affected subtree after each step. If heading structure changes, the executor may lift the refresh to a parent scope. For a single operator $o$, let

$$\text{Refresh}(o) := \{\, n \in N : n \text{ is the root of a refresh scope induced by some } \text{EDIT}(\cdot, \cdot, \emptyset) \text{ in that step} \,\},$$

and write $\text{Subtree}(\text{Refresh}(o))$ for all nodes in those refreshed scopes in the *pre-step* hierarchy.

**Payload footprint.** For a single operator $o$, define the payload footprint

$$
\begin{aligned}
F_{\text{payload}}(o) := {}& \text{Targets}_o(\text{EDIT}) \ \cup\ \text{Subtree}(\text{Refresh}(o)) \\
& \cup\ \text{Parents}_{\text{payload}}(o) \ \cup\ \text{Inserted}(o) \ \cup\ \text{Deleted}(o) \ \cup\ \text{Adjacent}(o),
\end{aligned}
$$

This set contains the nodes whose local payload may be created, removed, or rewritten by the step. It is separate from the structural footprint: a pure MOVE changes parent–child links and sibling order, but it does not put the moved subtree in $F_{\text{payload}}$ merely because the subtree moved. The moved subtree's local payload is instead required to be copied byte-for-byte by the subtree-preservation invariant. The term $\text{Adjacent}(o)$ tracks adjacency primarily through the boundary gap nodes (inter-paragraph whitespace) at the source and destination boundaries of each MOVE/INSERT/DELETE in that step. Because whitespace is modeled as payload, those gap nodes are payload-touching by construction. The boundary leaf nodes on each side are included only as a conservative halo because separator insertion can touch their local payload. These nodes may be handled as *soft* under optional normalization policies. Here $\text{Inserted}(o)$ includes nodes created by explicit INSERT *or* by refresh-based re-extraction, and $\text{Deleted}(o)$ includes nodes removed by DELETE *or* by refresh-based re-extraction. We define

$$
\begin{aligned}
\text{Parents}_{\text{payload}}(o) := {}& \text{Parents}_o(\text{MOVE} \cup \text{INSERT} \cup \text{DELETE}) \\
& \cup\ \text{Ancestors}(\text{Parents}_o(\text{MOVE} \cup \text{INSERT} \cup \text{DELETE}))\,,
\end{aligned}
$$

where $\text{Parents}_o(\cdot)$ denotes parents whose child ordering or local delimiters may change because of MOVE/ INSERT/DELETE in that step, and $\text{Ancestors}(S)$ denotes all ancestors of nodes in $S$. This expansion accounts for local payload owned by parents (inter-child whitespace and delimiters): reordering, inserting, or deleting children can change that local payload even when child content is unchanged. This makes the payload region explicit: any payload side effects (creation, modification, deletion) are confined to $F_{\text{payload}}(o)$ for each step. Some boundary nodes inside the footprint may still be payload-identical in a particular execution; the claim is that no node outside the footprint may change payload.

**Structural footprint.** Define the structural footprint for a single step as

$$
\begin{aligned}
F_{\text{struct}}(o) := \{\, n \in N \mid {}& \text{node } n\text{'s label/parent/children/sibling position} \\
& \text{or existence changes under } R \xrightarrow{o} R' \,\},
\end{aligned}
$$

which is computed by simulating the step and collecting nodes whose *structural signature* changes (plus insertions/deletions). For intuition, a conservative superset of $F_{\text{struct}}(o)$ is given by the union of $\text{Targets}_o$, $\text{Parents}_o$, $\text{Subtree}(\text{Refresh}(o))$, $\text{Inserted}(o)$, and $\text{Deleted}(o)$; this superset may also include $\text{Anchors}_o$ when anchors are treated as "touched" for access control. $F_{\text{payload}}(o)$ overlaps $F_{\text{struct}}(o)$ but may include nodes whose structure is unchanged (boundary siblings via the separator halo or parents with local delimiters). Structure-only operations can therefore have nonempty structural footprints while preserving the payload and rendered content of the moved subtrees; their payload footprint is limited to any boundary separator or delimiter payload that may be touched. Cross-parent moves change the child lists of the source and destination parents; any ancestor whose own child list or delimiter ownership must change is included as well. We therefore assume the intended target set $T$ is *closed* under these structural dependencies. The executor computes $F_{\text{struct}}(o_i)$ per step and either checks $F_{\text{struct}}(o_i) \subseteq T$ when a planner-provided $T$ is available, or sets $T := \bigcup_i F_{\text{struct}}(o_i)$ when absent.

**Footprint callout.** The executor enforces $F_{\text{struct}}(o_i)$ as the minimal region where *structure* may change for each step, and the *payload* side-effect claim is scoped to $F_{\text{payload}}(o_i)$. Spans may be reindexed deterministically due to payload-length shifts or refreshes. Local payload outside $F_{\text{payload}}(o_i)$ is unchanged for that step. Label changes are structural and therefore included in $F_{\text{struct}}$. Structure-only operators preserve the rendered content of moved subtrees by the subtree-preservation invariant. The rendered content of a parent can change when children move even if the local payload does not. The executor checks payload preservation per step by reconciling entity identifiers across pre/post hierarchies and comparing a *local payload signature* (text outside child spans) for entities outside $F_{\text{payload}}(o_i)$, so span reindexing is permitted while payload changes are rejected. This local payload corresponds to $p(n)$; the full rendered content can change when children move and is therefore not used for non-interference checks. The executor supports a configurable non-interference policy. Policy A (default) enforces strict byte-identical payloads outside $F_{\text{payload}}(o_i)$ on the byte-preserving hierarchy representation. Policy variants can permit whitespace-only separator repair, but only by making the repaired separator part of $F_{\text{payload}}(o_i)$. These variants are not assumed by Theorem 5 unless explicitly stated.

**Whitespace halo.** The whitespace halo handles an uncommon boundary case in byte-preserving prose edits. Most paragraphs already carry enough surrounding whitespace for a move or insertion to render as intended. The issue arises when an edit changes adjacency at a boundary where one side lacks an explicit separator. For example, a paragraph moved from the end of a document may have no trailing blank line; placing it immediately before another paragraph can make the two paragraphs render as one node under literal byte preservation. The tradeoff is therefore explicit. Strict byte preservation keeps the original whitespace exactly, but may accept this unintended boundary merge. Separator repair instead treats the missing separator as a small in-footprint payload change, represented by the whitespace halo, so the neighboring semantic nodes remain separate. The halo is limited to the gap nodes between neighboring semantic siblings at the edit boundary and the boundary leaf nodes of those neighbors. Nodes outside that small region remain subject to the usual payload-preservation check. Appendix F gives examples and the concrete footprint procedure.

**Entity reconciliation and scoped refresh.** When refresh-based execution re-extracts a hierarchy, the executor computes a reconciliation map between pre- and post-refresh nodes. Reconciliation is structure-first (parent/position anchors and spans are primary signals) with payload similarity used only as a tie-breaker. Outside the payload footprint, reconciliation must be total and unambiguous; otherwise execution fails (fail-closed). Inside the footprint, unmatched nodes are permitted and receive fresh entity identifiers. Merge and split events create new entity identifiers and record lineage edges, so continuity is preserved without reusing prior identities. When a plan allows a full-document edit ($T=N$), the reconciliation requirement becomes vacuous because every node lies inside the footprint.

**Remark (Why closure is necessary).** The closure condition on $T$ is not an extra "weakening" of the claim; it captures the structural dependencies induced by an operation. For example, a cross-parent MOVE that relocates a paragraph entails (i) updating the source parent's child list and (ii) updating the destination parent's child list. Those parents are structurally touched; ancestors are included only when the represen-

tation makes their structural lists or delimiter ownership part of the update. Their local payloads may remain identical. If the representation stores boundary delimiters as parent payload, or if separator repair is enabled, the relevant parent-owned delimiter or adjacent gap payload is included in $F_{\text{payload}}$. Declaring the structurally touched parents in $T$ isolates the structural changes while preserving the core claim: every node outside $\bigcup_i F_{\text{payload}}(o_i)$ remains payload-identical under the model assumptions.

**Invariant 1 (Payload Non-Interference).** For a single step $R_{i-1} \xrightarrow{o_i} R_i$, if a node $n \notin F_{\text{payload}}(o_i)$, then its payload is unchanged in that step:

$$n \notin F_{\text{payload}}(o_i) \;\Rightarrow\; p_{R_i}(n) = p_{R_{i-1}}(n).$$

**Invariant 1a (Rendered-Span Note).** Rendered content is derived from payload and children, so span indices can shift deterministically when payload lengths change; nodes may be outside $F_{\text{struct}}$ even if their spans shift. Payload preservation claims are limited to nodes outside $F_{\text{payload}}(o_i)$ for each step and to moved subtrees by the subtree-preservation invariant. Rendered spans may change under edits or moves without violating payload non-interference.

**Invariant 2 (Subtree Preservation for Structural Edits).** If an operator rearranges nodes but does not specify new payload, then the payload of any moved subtree is preserved exactly:

$$o_i \in \{\text{Move}(\pi), \text{Move}(\cdot)\} \;\Rightarrow\; p_{R_i}(n) = p_{R_{i-1}}(n) \quad \text{for all } n \text{ in the moved subtree of } o_i.$$

Equivalently, the rendered content of the moved subtree is identical and only its position in the document changes.

**Invariant 3 (Structural Soundness).** For every execution, the resulting hierarchy remains well-formed: nesting, acyclicity, and non-overlap constraints hold under the stated assumptions. The executor is designed to reject or repair any operation that would violate structural integrity.

**Discussion.** The executor enforces these invariants for plans that pass validation, independent of how those plans were sampled. A planner can still emit malformed or unsafe plans; those cases are reliability failures of the planning interface, and the executor either rejects them or the theorem's assumptions do not apply.

**Bounded model-checking complement.** In addition to the theorem-level proofs in this section, we model-check a bounded TLA+ abstraction of the orchestration loop and worker-output guardrails. The checked properties are FS-INV-001 (NoSideEffects), FS-INV-002 (NoIntroducedArtifacts), FS-INV-003 (RetryBoundedness), and FS-INV-004 (TerminationUnderBounds). Appendix G summarizes the checked bounds and explicit out-of-scope realism assumptions.

**Lemma 1** (Preservation). *Let $R$ be a well-formed hierarchy, and let $o$ be a single symbolic operator satisfying the typing rules of Section 2.4. If*

$$R \xrightarrow{o} R',$$

*then $R'$ is also a well-formed hierarchy (nesting, acyclicity, and non-overlap).*

*Proof.* By typing, $o$ belongs to the edit algebra $\mathcal{O}$ and satisfies the structural preconditions for its operator type:

- Move$(\pi)$ implements a pure permutation of section buffers; moved subtree bytes are preserved, and spans may be recomputed deterministically, but nesting and non-overlap are preserved by construction.

- Move$(n, \text{pos})$ relocates the subtree rooted at $n$. The executor updates parent/child links and re-renders the document from leaf payloads before spans are reindexed. The executor additionally rejects cycles (e.g., moving a node into its own subtree) and rejects any move that would violate non-overlap. Hence well-formedness is preserved.

- EDIT$(s_i, r, \theta)$ modifies the content of a crop bracketed entirely within $s(s_i)$; when the edit can change internal structure (e.g., $\theta$ is empty and the replacement may split or merge leaves), the affected subtree (or a lifted parent scope) is re-extracted deterministically and therefore satisfies well-formedness by construction. Otherwise, the executor does not change any spans or structural relations, so the hierarchy remains well-formed.

- INSERT$(\text{content}, \ell, \text{pos})$ adds a new node with a fresh identifier whose span lies within its parent span (when known) and does not overlap siblings. Affected child lists are updated deterministically, and existing node spans are reindexed from the render order, so well-formedness is preserved.

- DELETE$(n)$ removes a subtree; the executor updates parent–child links to maintain structural validity.

- ANNOTATE$(n, \ell)$ updates a node's label or metadata without changing its payload or child relations; this changes the structural signature but preserves well-formedness.

Thus no operator in $\mathcal{O}$ can introduce overlapping spans, cyclic parentage, or violations of nesting. Hence $R'$ is well-formed. □

**Lemma 2** (One-step payload locality)**.** *Let $R$ be a well-formed hierarchy and let $o$ be a well-typed operator that succeeds with accepted replacement $y$ (empty when no worker output is used), producing $R' = E(R, (o), y)$. Interpret $F^R_{\text{payload}}(o)$ as the step's allowed payload-touch set, determined by the operator, the pre-step hierarchy, and any deterministic refresh scope or reconciliation performed by the executor rather than by a post-hoc diff over arbitrary changes. Then every persistent node identity outside that set has identical local payload in $R$ and $R'$:*

$$n \notin F^R_{\text{payload}}(o) \;\Rightarrow\; p_{R'}(n) = p_R(n).$$

*If execution requires refresh or reconciliation, this conclusion holds only when the deterministic refresh accepts the step; otherwise the executor fails closed before returning $R'$.*

*Proof.* The claim follows from the executor interface and the definition of $F^R_{\text{payload}}(o)$. Each operator exposes only a bounded payload-write surface. EDIT can write only the selected target, crop, or refreshed subtree, all of which are included in $F^R_{\text{payload}}(o)$. INSERT and DELETE can create or remove nodes and may alter parent-owned delimiters or adjacent separators; inserted/deleted nodes, payload-owning parents, and adjacent halo nodes are included in the footprint. MOVE changes structure and possibly boundary separators, but does not rewrite the moved subtree's payload; any source/destination delimiter or separator payload that may be touched is included in the footprint, and moved-subtree payloads are protected by the subtree-preservation invariant. ANNOTATE changes labels or metadata, not payload.

For any node identity outside $F^R_{\text{payload}}(o)$, the executor therefore has no successful payload write path. If a refresh is required, reconciliation outside the footprint must be total and payload-identical; otherwise the step fails closed rather than returning $R'$. Hence any successful returned hierarchy has identical local payload for every persistent node outside $F^R_{\text{payload}}(o)$. □

**Lemma 3** (Progress)**.** *Let $R$ be a well-formed hierarchy, and let $o$ be a well-typed operator. Then either:*

1. *$R \xrightarrow{o} R'$ for a unique next hierarchy $R'$, or*

2. *the executor returns an explicit fail-closed error before applying the step.*

*When worker calls succeed and their accepted outputs are parseable by the deterministic extractors, the step $R \xrightarrow{o} R'$ is total and deterministic; otherwise the executor reports a worker-output error without silently applying a partial edit.*

*Proof.* Well-typed operators fall into one of five cases.

*Case 1:* $o = \text{MOVE}(\pi)$. A valid permutation can be applied by reassembling section buffers. No stochasticity is introduced by this operator in the formal model, so $R'$ exists uniquely.

*Case 2:* $o = \text{EDIT}(s_i, r, \theta)$. Typing stipulates $s_i$ exists and $\theta$ (if provided) is within its span. Thus the executor can: (i) extract the needed content, (ii) call the worker LLM (or use provided replacement content), (iii) splice the result, yielding a unique $R'$ when the worker succeeds and its output is parseable. If the edit can change internal structure (e.g., $\theta$ is empty and the replacement may split or merge leaves), subtree (or lifted parent-scope) re-extraction is deterministic under the same parseability assumption.

*Case 3:* $o = \text{INSERT}(\text{content}, \ell, \text{pos})$. Typing stipulates pos is valid. The executor can create the new node and insert it, yielding a unique $R'$.

*Case 4:* $o = \text{DELETE}(n)$. Typing stipulates $n \in N$. The executor can remove the subtree rooted at $n$, yielding a unique $R'$.

*Case 5:* $o = \text{ANNOTATE}(n, \ell)$. Typing stipulates $n \in N$ and $\ell$ is a valid label. The executor updates the node's label/metadata without modifying payload or parent/child relations; the structural signature changes only in the label, yielding a unique $R'$.

In all successful cases, the executor defines a unique next state. In the remaining cases, validation or worker-output checking returns an explicit error before applying the step. Hence a well-typed single step either progresses to a unique next hierarchy or fails closed rather than becoming stuck. $\square$

**Corollary 4** (Type Soundness for Symbolic Plans)**.** *Repeated execution of a well-typed symbolic plan:*

$$R_0 \xrightarrow{o_1} R_1 \xrightarrow{o_2} \ldots \xrightarrow{o_k} R_k$$

*preserves well-formedness at every successful step. Under the typing assumptions and accepted worker-output contracts, execution does not get stuck; detectable contract violations return explicit fail-closed errors before applying the violating step.*

### 4.3 Deterministic Safety Theorem

The invariants above yield our primary correctness result: if the input has been correctly extracted into the hierarchy used by the executor, the plan is well-typed and footprint-contained, and accepted worker outputs satisfy the executor interface, then execution is payload-side-effect-free and structurally well-formed in the formal model.

**Theorem 5** (Deterministic Safety)**.** *Let $X$ be an input document, let $R = (N, E, \ell, s, p)$ be a well-formed hierarchy that correctly extracts $X$ under the canonicalization policy of Section 2, let $T$ be the intended structural target set over node identities in the simulated trace, let $P = (o_1, \ldots, o_k)$ be a symbolic plan, and let $Y = (y_1, \ldots, y_k)$ be the accepted local replacements used by worker-generation steps (empty for non-worker steps). Suppose $\text{Good}(P, R, Y)$ holds, where validation simulates execution over intermediate hierarchies $R_0 = R, \ldots, R_k$ and:*

(i) *each $o_i$ belongs to $\mathcal{O}$, is well-typed for the pre-step hierarchy $R_{i-1}$ produced by the simulated prefix execution (Section 2.4), and satisfies its structural preconditions on $R_{i-1}$ (including span containment and non-overlap for* INSERT*),*

(ii) *$F_{\text{struct}}^{R_{i-1}}(o_i) \subseteq T$ for every step (equivalently, $\bigcup_i F_{\text{struct}}^{R_{i-1}}(o_i) \subseteq T$ over the simulated trace), and*

(iii) *every worker output used by a* EDIT *step satisfies the executor interface for the plan-determined local worker request, and any required refresh/reconciliation check accepts the step.*

*Then the executor output $R' = E(R, P, Y)$ is payload-side-effect-free in the formal model: for every step $R_{i-1} \xrightarrow{o_i, y_i} R_i$, all nodes $n \notin F_{\text{payload}}^{R_{i-1}}(o_i)$ retain identical payload in that step, structure-only moves preserve the rendered content of moved subtrees by the subtree-preservation invariant, and every $R_i$ is structurally*

well-formed. *Consequently, any persistent node identity outside the stepwise union $\bigcup_i F_{\text{payload}}^{R_{i-1}}(o_i)$ retains identical payload in $R'$. For full-document rewrites, choose $T$ to include the whole evolving node universe, in which case the side-effect claim becomes vacuous but the same execution semantics apply.*

**Composition.** Because execution is deterministic for a fixed plan and fixed accepted replacements, sequential composition agrees with batching when the plan is executed as a single fixed sequence: for any two plans $P_1, P_2$ that are each well-typed and footprint-contained, with accepted replacements $Y_1, Y_2$, $E(R, P_1 \parallel P_2, Y_1 \parallel Y_2) = E(E(R, P_1, Y_1), P_2, Y_2)$ provided the combined structural footprint respects closure and the plan remains well-typed on the intermediate hierarchy. This is the case when the executor validates the full plan and does not trigger replanning. This justifies emitting compound edits either as a single plan or as a series of calls without changing observable outcomes once accepted local replacements are fixed. When the runtime replans between steps, the planning stage emits a new plan conditioned on the updated hierarchy, so the composition statement applies per plan segment rather than to the entire replan loop.

*Proof.* We view execution as a small-step relation on hierarchies: for a single operator $o$ and accepted local replacement $y$ (empty when no worker output is used), write

$$R \xrightarrow{o,y} R' \quad \text{iff} \quad R' = E(R, (o), y).$$

By definition of $E$, multi-step execution of a plan $P = (o_1, \ldots, o_k)$ is the reflexive, transitive closure of this relation:

$$R \xrightarrow{o_1, y_1} R_1 \xrightarrow{o_2, y_2} \ldots \xrightarrow{o_k, y_k} R_k,$$

with $R_k = E(R, P, Y)$.

We prove that, under $\text{Good}(P, R, Y)$, every step preserves (i) structural well-formedness and (ii) payload identity of all nodes outside the payload footprint; the theorem then follows by composition of steps.

*Step-wise preservation.* Fix an intermediate hierarchy $R_j = (N, E, \ell, s, p)$ and the next operator $o_{j+1}$ satisfying the clauses of $\text{Good}(P, R, Y)$ on $R_j$:

(a) $o_{j+1}$ belongs to $\mathcal{O}$, is well-typed for $R_j$, and satisfies its structural preconditions,

(b) $F_{\text{struct}}^{R_j}(o_{j+1}) \subseteq T$, and

(c) any worker output for $o_{j+1}$ satisfies the executor interface.

Let $R_{j+1}$ be the result of applying $o_{j+1}$ with accepted replacement $y_{j+1}$, i.e. $R_j \xrightarrow{o_{j+1}, y_{j+1}} R_{j+1}$.

By (a), the typing rules of Section 2.4 and the structural preconditions imply that the executor applies $o_{j+1}$ without violating any local structural constraints. Lemma 1 therefore implies that $R_{j+1}$ is well-formed whenever $R_j$ is well-formed.

By Lemma 2, for all $n \notin F_{\text{payload}}^{R_j}(o_{j+1})$,

$$p_{R_{j+1}}(n) = p_{R_j}(n).$$

Clause (b) ensures the structural changes for $o_{j+1}$ are confined to $T$. For MOVE operators, the subtree-preservation invariant shows that each moved subtree preserves its rendered content; for a root-level permutation this applies to each moved root subtree.

*Global preservation.* We now compose these step-wise properties along the unique execution trace induced by $P$. Write

$$R_0 = R, \quad R_{j+1} = E(R_j, (o_{j+1}), y_{j+1}), \quad R_k = E(R, P, Y).$$

Since $R_0$ is well-formed by assumption, repeated application of Lemma 1 shows that every $R_j$ is well-formed, in particular $R_k = R'$. Likewise, repeated application of Lemma 2 shows that for all persistent node identities $n \notin \bigcup_i F_{\text{payload}}^{R_{i-1}}(o_i)$ and all $j$, the payload of $n$ is identical in $R_j$ and $R_0$.

Therefore, in the final hierarchy $R'$, every persistent node identity $n \notin \bigcup_i F_{\text{payload}}^{R_{i-1}}(o_i)$ is payload-identical relative to $R$, and the hierarchy remains well-formed. This is precisely the payload-side-effect-free property claimed in the theorem. □

**Scope of the deterministic result.** Theorem 5 is a pointwise executor guarantee, not a probabilistic claim about an LLM planner. It applies after the document has been correctly extracted into the hierarchy used by the executor, after the planner has emitted a well-typed symbolic plan with footprint closure, and after worker outputs satisfy the executor interface. Failures in extraction, planning, or worker-output conformance are therefore empirical reliability questions rather than consequences of the theorem. If a plan is malformed, refers to nonexistent nodes, or violates a worker-output contract that the implementation can detect, the executor rejects the step and performs no edit.

## 5 Experiments

We evaluate Functional Safety (FS) on tasks chosen to separate local editing, structural reorganization, and multi-location bookkeeping: targeted paragraph edits, contextual insertion, code visibility refactors, long-form section relocation, paragraph consolidation, LaTeX micro-edits, and multi-file refactors. The main result is that, holding the LLM fixed, changing the editing architecture around the model can substantially change outcomes. FS improves observed reliability over the standard ReAct+`apply_patch` workflow on structured editing tasks by giving the model an explicit hierarchy, typed edit operators, and executor-side locality checks rather than relying on context-matching patches alone.

The single-document experiments compare three editing paradigms: (i) a *flat autoregressive baseline* that rewrites full document spans, (ii) a *ReAct+`apply_patch` tool agent* with navigation, search, and context-matching patch actions, and (iii) the *hierarchy-aware Functional Safety planner–executor*. ReAct+`apply_patch` is strongest on local prose edits and paragraph consolidation, but becomes brittle when edits require coordinated changes across distant document regions, code/prose boundaries, or linked LaTeX structures such as labels, references, and citations.

The multi-file refactors test the same architectural claim at repository scale. The hierarchy-aware system plans over extracted code spans and materializes files from those spans, rather than asking the model to regenerate each target file as free-form text. This is why a general-purpose API model can be competitive with frontier CLI editors in this setting: on the 13-file `compute` refactor, `gpt-4o` FS, Codex CLI (with GPT-5.2-Codex), and Claude Code (with Claude Opus 4.5) all reach 100% test pass, but `gpt-4o` FS averages 4.9 s versus 119 s and 1124 s. On the 49-file `vector` refactor, `gpt-4o` FS reaches 98% test pass versus 100% for Codex CLI and 97% for Claude Code, while averaging 23 s versus 564 s and 1257 s. Exact identity and import metrics vary by task; Tables 4–5 report the main comparison. Stress tests (pure permutation and compound multi-operation edits) are summarized in Appendix E.

Table 1: Experimental overview. Each row represents a main experiment with its content type, scale, task, and corresponding results.

| Experiment | Content | Scale | Task | Tables |
|---|---|---|---|---|
| Targeted Paragraph Editing | Policy documents | ≈2k chars | Single paragraph fix | 2 |
| Contextual Insert | Policy documents | ≈2k chars | Insert summary with context | 2 |
| Code Visibility Refactor | Python module | ≈1.7k–3.1k chars | Group private methods + fix typo | 2 |
| Long-Form Section Relocation | Policy briefs | 10k, 20k chars | Move section + fix typo | 2 |
| Paragraph Consolidation | Policy briefs | 10k, 20k chars | Merge two paragraphs | 2 |
| LaTeX Micro-Edits | LaTeX (book) | 10k, 20k chars | Fix math / \ref / \cite typos | 2 |
| Code Multi-File Refactor (Compute) | Python (package) | 13 files | Split monolith into package tree | 4 |
| Code Multi-File Refactor (Vector) | Python (package) | 49 files | Split monolith into package tree | 5 |
| Code Multi-File Refactor (Imports-top) | Python (package) | 13, 49 files | Split monolith with consolidated imports | 29, 30 |
| Code Multi-File Refactor (Open, relaxed) | Python (package) | 13, 49 files | Open-ended split into package tree | 31, 33 |
| Code Multi-File Refactor (Open, compat) | Python (package) | 13, 49 files | Open-ended split with import-path compatibility | 32, 34 |
| LaTeX Multi-File Refactor | LaTeX (book) | 5 files | Split inline chapters into `chapters/` | 6 |
| LaTeX Multi-File Refactor (Appendix) | LaTeX (book) | 4 files | Split inline appendix with cross-refs | 7 |

Table 2: Summary of remote-model results split by model. All Succ./SE columns use whitespace-normalized (WS) metrics. For FS and ReAct+`apply_patch`, the per-task tables in Appendix E show that WS and strict (byte-level) rates coincide in every row, so the FS and ReAct+patch columns can be read as strict as well. The Flat baseline can be content-correct under WS on short prose tasks but never preserves bytes outside the target, so its WS and strict numbers differ on those tasks. All rates are proportions with 95% Wilson confidence intervals.

| Task Family | Model | Flat Succ. | Flat SE | ReAct+patch Succ. | ReAct+patch SE | FS Succ. | FS SE | N |
|---|---|---|---|---|---|---|---|---|
| Targeted Paragraph Editing | gpt-4.1 | 1.00 [0.93, 1.00] | 0.00 [0.00, 0.07] | 1.00 [0.93, 1.00] | 0.00 [0.00, 0.07] | 1.00 [0.93, 1.00] | 0.00 [0.00, 0.07] | 50 |
| | grok-4-1-fast-reasoning | 0.70 [0.56, 0.81] | 0.30 [0.19, 0.44] | 0.96 [0.87, 0.99] | 0.06 [0.02, 0.16] | 0.70 [0.56, 0.81] | 0.00 [0.00, 0.07] | 50 |
| Contextual Insert | gpt-4.1 | 1.00 [0.93, 1.00] | 0.00 [0.00, 0.07] | 0.96 [0.87, 0.99] | 0.04 [0.01, 0.13] | 0.82 [0.69, 0.90] | 0.00 [0.00, 0.07] | 50 |
| | grok-4-1-fast-reasoning | 0.70 [0.56, 0.81] | 0.30 [0.19, 0.44] | 0.60 [0.46, 0.72] | 0.38 [0.26, 0.52] | 0.70 [0.56, 0.81] | 0.00 [0.00, 0.07] | 50 |
| Code Visibility Refactor | gpt-4.1 | 0.00 [0.00, 0.04] | 1.00 [0.96, 1.00] | 0.01 [0.00, 0.05] | 0.41 [0.32, 0.51] | 1.00 [0.96, 1.00] | 0.00 [0.00, 0.04] | 100 |
| | grok-4-1-fast-reasoning | 0.04 [0.02, 0.10] | 0.96 [0.90, 0.98] | 0.02 [0.01, 0.07] | 0.17 [0.11, 0.26] | 0.70 [0.60, 0.78] | 0.00 [0.00, 0.04] | 100 |
| Long-Form Section Relocation | gpt-4.1 | 0.00 [0.00, 0.04] | 1.00 [0.96, 1.00] | 0.00 [0.00, 0.04] | 0.93 [0.86, 0.97] | 1.00 [0.96, 1.00] | 0.00 [0.00, 0.04] | 100 |
| | grok-4-1-fast-reasoning | 0.00 [0.00, 0.04] | 1.00 [0.96, 1.00] | 0.23 [0.16, 0.32] | 0.47 [0.38, 0.57] | 0.70 [0.60, 0.78] | 0.00 [0.00, 0.04] | 100 |
| Paragraph Consolidation | gpt-4.1 | 0.00 [0.00, 0.04] | 1.00 [0.96, 1.00] | 0.83 [0.74, 0.89] | 0.03 [0.01, 0.08] | 1.00 [0.96, 1.00] | 0.00 [0.00, 0.04] | 100 |
| | grok-4-1-fast-reasoning | 0.00 [0.00, 0.04] | 1.00 [0.96, 1.00] | 0.86 [0.78, 0.91] | 0.00 [0.00, 0.04] | 0.70 [0.60, 0.78] | 0.00 [0.00, 0.04] | 100 |
| LaTeX Micro-Edits | gpt-4.1 | 0.00 [0.00, 0.01] | 1.00 [0.99, 1.00] | 0.52 [0.46, 0.57] | 0.48 [0.43, 0.54] | 1.00 [0.99, 1.00] | 0.00 [0.00, 0.01] | 300 |
| | grok-4-1-fast-reasoning | 0.00 [0.00, 0.01] | 1.00 [0.99, 1.00] | 0.94 [0.90, 0.96] | 0.06 [0.04, 0.10] | 0.70 [0.65, 0.75] | 0.00 [0.00, 0.01] | 300 |

**Experimental Overview.** Table 2 is the main comparison for the six single-document task families. These tasks use deterministic document instances with repeated model calls: targeted paragraph edits and contextual insertions use ≈2k-character policy documents; code-visibility refactors use 1.7k–3.1k-character Python classes; long-form relocation and paragraph consolidation use 10k- and 20k-character policy briefs; and LaTeX micro-edits use book-style generated briefs with labels, citations, and math lines. The flat baseline is included as a lower bound for full-span regeneration, but the central comparison is between ReAct+`apply_patch` and FS. The ReAct baselines are described in Section 5.5, with per-condition detail tables in Appendix E.

Table 1 summarizes the experimental progression. Each experiment isolates a different dimension of difficulty: document scale (2k to 20k characters), content type (prose vs. code), and task complexity (single-target edits vs. compound structural-plus-semantic operations).

**Corpus provenance and repetitions.** A document instance is a deterministic benchmark input keyed by a document ID; repetitions vary only the model call, not the source text. Unless a caption states otherwise, the remote single-document rows use 10 document IDs with five repetitions per document. Targeted paragraph and contextual-insert tasks use synthetic policy documents generated from seeded section and paragraph templates. Code-visibility tasks use seeded Python modules with generated classes, methods, and target typos. Long-form relocation and consolidation tasks use a policy-brief generator with a fixed section taxonomy but document-ID-specific paragraph text, padding, distractors, and target placements. LaTeX micro-edits render the same generated brief family into book-style LaTeX instances with document-

ID-specific chapters, labels, citations, and math lines. Multi-file refactors use deterministic fixtures from the repository's Python and LaTeX fixture trees.

Remote model calls use provider-default decoding settings, and local-model calls use $T = 0.3$ unless otherwise stated.

For single-document tasks, *Success Rate* denotes the fraction of trials that satisfy the task oracle and have no forbidden side effects under the reported metric. *Side-Effect Rate* denotes the fraction of trials with changes outside the task-defined target region (not the planner footprint). We report strict byte-level variants and whitespace-normalized (WS) variants; the no-side-effect rate is 1 minus the corresponding side-effect rate. Parser, plan, or worker-output rejections are counted as task failures, but not as side-effect events unless an output is applied and violates the side-effect oracle. For rate estimates we report 95% Wilson confidence intervals. In generated flat/FS detail tables, the $p$ columns compare rows within the same task/size/model condition using paired McNemar exact tests when matched discordant counts are available or inferable from degenerate equal-size marginals, and a two-proportion $z$ test otherwise. These exploratory $p$ values are unadjusted for multiple comparisons. ReAct comparators are described in Section 5.5.

The multi-file refactor tables use a different metric set. In those tables, *Test Pass* measures downstream unit-test behavior of a reconstructed codebase; it is not the same quantity as single-document strict success or no-side-effect rate. Byte identity, whitespace-normalized identity, AST identity, import/signature deltas, layout match, and test pass should be read as separate views of refactor quality rather than as one unified success metric.

**Hierarchy extraction evidence.** All structured-format experiments use host-side parsers or syntax-driven span assignment to build the hierarchy before planning. Because extraction quality is a safety-critical dependency, Appendix C reports two diagnostics. On four structured Markdown/LaTeX fixtures with 15 repetitions per model, LLM-first extraction returns every document but remains much slower and less span-reliable than host parsing: the LLM runs take 2.36 s (`gpt-4.1`) to 32.07 s (`grok-4-1-fast-reasoning`) on average versus 0.00012 s for host parsing, with overall section F1@0.9 of 0.21–0.46 and paragraph F1@0.9 of 0.38. We therefore use deterministic extraction whenever the input format supports it.

**Weakly structured input pilot.** For prose without reliable headings, the appendix reports a nine-paper pilot that uses LLM grouping only for sections while snapping paragraph spans to deterministic blank-line paragraph boundaries. This hierarchy-based paragraph edit achieves 9/9 exact-match edits with no outside diffs; a ReAct+`apply_patch` comparator on the same target paragraphs reaches 2/9, with eight of nine runs hitting the parse-error budget without applying any patch and the one run that did finish modifying 252 bytes outside the target paragraph. The result supports conservative paragraph anchoring as a useful fallback for weakly structured text, but it is not a broad reliability claim for arbitrary unstructured documents: if extraction groups the wrong paragraphs or loses byte-aligned spans, the formal executor guarantee no longer applies to the intended document units.

## 5.1 Single-Document Editing Results

Stress-test diagnostics are reported in Appendix E. For paragraph consolidation, success is defined as reducing the target section's paragraph count by exactly one.

Table 3: Compact local/open-model summary for the smaller tasks, reporting whitespace-normalized (WS) and strict counts. Rows are grouped by run budget: `llama3:8b`, `qwen2.5:14b`, and `gemma3:27b` use 15 runs per task, while `qwen3.5:27b` uses 50 runs per task over 10 document IDs. Full per-task Wilson intervals appear in Appendix B.

| Model | Conditions | Flat WS Succ. | Flat Strict Succ. | Flat WS SE | Flat Strict SE | FS WS Succ. | FS Strict Succ. | FS WS SE | FS Strict SE |
|---|---|---|---|---|---|---|---|---|---|
| *15-run local-model rows* | | | | | | | | | |
| llama3:8b | $3 \times 15$ | 0/45 | 0/45 | 45/45 | 45/45 | 45/45 | 45/45 | 0/45 | 0/45 |
| qwen2.5:14b | $3 \times 15$ | 15/45 | 0/45 | 30/45 | 45/45 | 45/45 | 45/45 | 0/45 | 0/45 |
| gemma3:27b | $3 \times 15$ | 30/45 | 0/45 | 15/45 | 45/45 | 45/45 | 45/45 | 0/45 | 0/45 |
| *50-run qwen3.5 row* | | | | | | | | | |
| qwen3.5:27b | $3 \times 50$ | 100/150 | 0/150 | 50/150 | 150/150 | 150/150 | 150/150 | 0/150 | 0/150 |

The local/open-model rows use two run budgets: 15 runs per task for `llama3:8b`, `qwen2.5:14b`, and `gemma3:27b`, and 50 runs per task for `qwen3.5:27b`. Across both budgets, all four evaluated local models solve the three smaller tasks under FS with zero observed strict side effects: 45/45 strict success for each 15-run model and 150/150 for `qwen3.5:27b`. The flat baseline has zero strict success in these rows because full-span rewrites change untargeted regions; its WS success is higher for some models, showing that some requested edits survive after formatting differences are ignored. Appendix B.1 reports a reasoning-mode sensitivity check, where additional reasoning budget increased latency and structured-output failures. These results are coverage for three smaller tasks, not evidence that the evaluated local models solve the longer relocation, LaTeX, or multi-file settings.

On local prose edits, ReAct+`apply_patch` is a strong baseline rather than a strawman. With `gpt-4.1` it reaches 100% success on targeted paragraph editing (0% side effects) and 83% on paragraph consolidation (3% side effects); with `grok-4-1-fast-reasoning` it reaches 96% and 86% on the same tasks. These are tasks where a context-matching patch can usually represent the needed text. FS with `gpt-4.1` matches or exceeds ReAct on both, reaching 100% on each with zero observed side effects; FS with `grok` is bounded at 70%, which is the same content-quality ceiling `grok`-Flat hits under WS on the short prose tasks. For the local/open-model runs in Table 3, all evaluated local models solve the three smaller tasks under FS with zero observed strict side effects.

The gap opens on tasks that require more bookkeeping than local text repair. On contextual insertion, `gpt-4.1` ReAct reaches 96% but `grok` ReAct drops to 60% with a 38% side-effect rate. Code visibility and long-form relocation are much less forgiving: ReAct+`apply_patch` has $\leq 2\%$ success on code visibility for both models, and 0% (`gpt-4.1`) or 23% (`grok`) success on section relocation, with high side-effect rates. In the corresponding FS rows, the executor applies typed hierarchy operations: method permutations preserve extracted method blocks while fixing the target typo, and section relocation delegates the permutation to the deterministic executor before applying one localized edit. FS with `gpt-4.1` reaches 100% on both code visibility and long-form relocation; FS with `grok` reaches 70% on each, matching its prose-task ceiling. Across these task families, FS has zero observed side effects in the remote rows for both models.

The recurring pattern is therefore simple: patch-style agents are competitive when the edit is local and easy to express as a small patch, but the representation becomes brittle when success depends on maintaining an explicit document structure. FS is designed for the latter regime: the model chooses or fills a bounded symbolic edit, while the executor controls where bytes may change.

## 5.2 LaTeX Micro-Edits

We evaluate localized edits on book-style LaTeX documents with chapter-level hierarchy. Each task targets a single line-level fix—an equation symbol, a cross-reference label plus its matching `\ref`, or a citation key inside `\cite`—and requires all other content to remain byte-identical. Table 2 summarizes the aggregate result. In these runs, flat baselines have 0/50 strict success and strict side-effect rate = 1.00 in each remote model/size/task row. The ReAct+`apply_patch` agent shows a task-dependent gradient: simple math-symbol fixes succeed reliably (98–100%) with low side-effect rates, but cross-reference updates requiring coordinated changes at two locations (`\label` and matching `\ref`) show strongly model-dependent results (`grok` 88–98% vs. `gpt-4.1` 24–26%), and citation-key fixes diverge similarly (`grok` 88–90% vs. `gpt-4.1` 24–38%).

Failures are not clean: the agent frequently edits the wrong location within the target chapter (e.g., replacing the wrong math symbol or label), which is detected as an intra-target side effect. Functional Safety has 0 observed side-effect rate in all reported micro-edit rows, with `gpt-4.1` reaching 50/50 strict success and `grok-4-1-fast-reasoning` reaching 35/50 in each reported row. Per-condition micro-edit tables and structural LaTeX diagnostics are reported in Appendix E.

## 5.3 Code Multi-File Refactor

We evaluate two long-running, multi-file Python refactors that split a monolithic file back into its original package layout: `compute` (13 files) and `vector` (49 files). Each monolith is created by concatenating the original package files; the task asks the model to reconstruct the original file tree and preserve public exports. These experiments compare Codex CLI and Claude Code CLI baselines to the hierarchical planner/executor (there is no flat full-span baseline for multi-file reconstruction). In the exact-layout cases, the target file paths are supplied to the planner; the model's task is to assign extracted definitions, classes, import blocks, and other code nodes from the monolith to those files. The executor then rebuilds each file from its assigned nodes, rather than asking the model to regenerate each target file as free-form text or to discover the refactor through a long patch-editing loop. The comparison is meant to test an architectural alternative to simply scaling agent intelligence: whether node-level decomposition of a monolith can achieve comparable behavioral correctness to frontier CLI editors, often with lower wall-clock time. Both CLI harnesses invoked the tools' configured defaults rather than passing explicit model IDs on the command line. The Codex rows were generated after OpenAI's December 18, 2025 GPT-5.2-Codex rollout to Codex surfaces (OpenAI, 2025), so we report the Codex CLI baseline as GPT-5.2-Codex. The Claude Code rows used a local default configured for Opus and were generated after Anthropic's November 24, 2025 Claude Opus 4.5 release (Anthropic, 2025), so we report the Claude Code baseline as Claude Opus 4.5. The table labels preserve the exact run identifiers used by the harness (e.g., `codex_cli_default_medium` and `claude_cli_reference`).

We report *Byte Id* (exact match), *Whitespace-normalized Id* (using the evaluator's line-preserving code policy: line endings are normalized and trailing horizontal whitespace/trailing blank lines are ignored, while line order and non-whitespace tokens remain significant), *AST Id* (AST-equivalence ignoring docstrings), *Sig* $\Delta$ (function signature diffs), *Import* $\Delta$ (missing/extra imports), *Repair Rounds* (mean number of planner repair loops per run), *Test Pass* (fraction of unit tests passing), and mean wall-clock time per run. Plans are validated; invalid plans produce no output files and are scored as missing. Runs were accumulated over multiple days on a fixed code snapshot. Tables 4 and 5 report the headline exact-layout comparisons; Appendix E reports the setup table plus imports-top and open-layout variants.

Table 4: Multi-file Python compute refactor (13 files). Metrics include structural equivalence, import/signature diffs, test pass rate, and timing.

| Approach | Runs | Failed | Layout Match | Byte Id | WS Id | TeX Id | AST Id | Sig Δ | Import Δ | Repair Rounds | Time (s) | Test Pass |
|---|---|---|---|---|---|---|---|---|---|---|---|---|
| claude_cli_reference | 5 | 0 | 5 | 0.92+/-0.05 | 0.92+/-0.05 | – | 0.92+/-0.05 | 2.4+/-3.4 | 1.4+/-1.7/1.4+/-3.1 | – | 1123.53+/-80.04 | 1.00+/-0.00 |
| codex_cli_default_medium | 5 | 0 | 5 | 0.08+/-0.00 | 0.08+/-0.00 | – | 1.00+/-0.00 | 0.0+/-0.0 | 0.0+/-0.0/0.0+/-0.0 | – | 119.00+/-10.31 | 1.00+/-0.00 |
| hier_gpt-4o | 5 | 0 | 5 | 0.85+/-0.00 | 0.85+/-0.00 | – | 0.85+/-0.00 | 0.0+/-0.0 | 53.0+/-0.0/49.0+/-0.0 | 1.2+/-0.4 | 4.87+/-1.20 | 1.00+/-0.00 |
| hier_gpt-5 | 5 | 0 | 5 | 0.94+/-0.08 | 0.94+/-0.08 | – | 0.94+/-0.08 | 0.0+/-0.0 | 21.2+/-29.0/19.6+/-26.8 | 1.0+/-0.0 | 29.23+/-3.61 | 1.00+/-0.00 |
| hier_grok-4-1-fast-reasoning | 5 | 0 | 5 | 0.85+/-0.00 | 0.85+/-0.00 | – | 0.85+/-0.00 | 0.0+/-0.0 | 53.0+/-0.0/49.0+/-0.0 | 1.0+/-0.0 | 25.57+/-1.75 | 1.00+/-0.00 |

Table 5: Multi-file Python vector refactor (49 files). Metrics include structural equivalence, import/signature diffs, test pass rate, and timing.

| Approach | Runs | Failed | Layout Match | Byte Id | WS Id | TeX Id | AST Id | Sig Δ | Import Δ | Repair Rounds | Time (s) | Test Pass |
|---|---|---|---|---|---|---|---|---|---|---|---|---|
| claude_cli_reference | 5 | 0 | 3 | 0.53+/-0.24 | 0.53+/-0.24 | – | 0.65+/-0.21 | 1.2+/-2.7 | 50.8+/-37.8/35.4+/-40.1 | – | 1256.93+/-730.61 | 0.97+/-0.01 |
| codex_cli_default_medium | 5 | 0 | 5 | 0.02+/-0.00 | 0.02+/-0.00 | – | 1.00+/-0.00 | 0.0+/-0.0 | 0.0+/-0.0/0.0+/-0.0 | – | 563.60+/-103.20 | 1.00+/-0.00 |
| hier_gpt-4o | 5 | 0 | 5 | 0.78+/-0.19 | 0.78+/-0.19 | – | 0.78+/-0.19 | 0.0+/-0.0 | 24.8+/-25.8/142.2+/-149.1 | 14.0+/-3.5 | 23.16+/-2.62 | 0.98+/-0.01 |
| hier_gpt-5 | 5 | 0 | 5 | 0.58+/-0.39 | 0.58+/-0.39 | – | 0.58+/-0.39 | 0.0+/-0.0 | 8.6+/-14.3/793.8+/-776.9 | 6.6+/-5.1 | 421.63+/-307.95 | 0.99+/-0.01 |
| hier_grok-4-1-fast-reasoning | 5 | 0 | 5 | 0.87+/-0.28 | 0.87+/-0.28 | – | 0.87+/-0.28 | 0.0+/-0.0 | 2.6+/-5.8/237.6+/-531.3 | 3.6+/-3.6 | 174.46+/-169.68 | 1.00+/-0.01 |

## 5.4 LaTeX Multi-File Refactor

We evaluate a multi-file refactor in which a book-style LaTeX document has chapter bodies inlined directly into `main.tex`, with each chapter preceded by its `\include` line. The task asks the model to reconstruct the original file tree under `chapters/`, restoring a standalone `main.tex` that contains only the preamble

and the `\include` directives. We report three identity metrics: *Byte Id* (exact file match), *Whitespace-normalized Id* (using the evaluator's line-preserving LaTeX policy: line endings are normalized and trailing horizontal whitespace/trailing blank lines are ignored), and *TeX Id* (that whitespace-normalized match plus normalization of `\include` and `\input` paths such as `\include{foo}` versus `\include{foo.tex}`), plus mean wall-clock time per run. We run five repetitions per approach and aggregate the means and standard deviations in Table 6.

Table 6: Multi-file LaTeX chapter refactor (5 runs). The TeX Id metric treats `\include` path suffixes as equivalent; timing is reported as mean wall-clock seconds.

| Approach | Runs | Failed | Layout Match | Byte Id | WS Id | TeX Id | Time (s) |
|---|---|---|---|---|---|---|---|
| claude_cli_reference | 5 | 0 | 5 | 0.93+/-0.09 | 0.93+/-0.09 | 0.93+/-0.09 | 73.50+/-8.63 |
| codex_cli_default_medium | 5 | 0 | 5 | 0.83+/-0.00 | 0.83+/-0.00 | 0.83+/-0.00 | 75.60+/-11.16 |
| hier_gpt-4o | 5 | 0 | 5 | 1.00+/-0.00 | 1.00+/-0.00 | 1.00+/-0.00 | 1.12+/-0.19 |
| hier_grok-4-1-fast-reasoning | 5 | 0 | 5 | 1.00+/-0.00 | 1.00+/-0.00 | 1.00+/-0.00 | 4.26+/-0.83 |

We additionally evaluate an appendix refactor where cross-file `\ref` links point from chapters to the appendix and back. This variant stresses whether refactoring preserves labels and references across file boundaries; the same identity metrics plus timing are reported in Table 7.

Table 7: Multi-file LaTeX appendix refactor with cross-file references (5 runs), reporting both correctness and timing.

| Approach | Runs | Failed | Layout Match | Byte Id | WS Id | TeX Id | Time (s) |
|---|---|---|---|---|---|---|---|
| claude_cli_reference | 5 | 0 | 5 | 1.00+/-0.00 | 1.00+/-0.00 | 1.00+/-0.00 | 57.53+/-9.93 |
| codex_cli_default_medium | 5 | 0 | 5 | 0.70+/-0.27 | 0.70+/-0.27 | 0.70+/-0.27 | 58.61+/-26.42 |
| hier_gpt-4o | 5 | 0 | 5 | 1.00+/-0.00 | 1.00+/-0.00 | 1.00+/-0.00 | 1.03+/-0.25 |
| hier_grok-4-1-fast-reasoning | 5 | 0 | 5 | 1.00+/-0.00 | 1.00+/-0.00 | 1.00+/-0.00 | 3.69+/-0.39 |

### 5.5 ReAct Tool-Agent Baselines

We evaluate ReAct-style tool agents as agentic editing baselines. The primary comparator is ReAct+`apply_patch`, which mirrors the context-matching patch workflow used by many CLI editing agents. A raw-offset variant is included as a diagnostic stress test for direct span editing, but it is not the headline comparator in the summary tables. At each step the agent receives the task instruction, the current document state, and tool schemas, then emits a JSON action envelope selecting one tool and its arguments. Invalid JSON triggers lightweight auto-repair (strip prose, fix trailing commas, normalize booleans, quote bare keys) before counting a parse error. Tool failures and parse errors each have bounded budgets; exceeding either terminates the episode.

We test two tool configurations:

- **ReAct+`apply_patch`.** Read-only tools (list_headings, search, search_lines, view, view_lines) plus a `apply_patch` tool that accepts unified diffs and applies them by matching context lines (`git apply`-style). This uses a 12-step budget and represents a common context-matching patch workflow rather than raw-offset editing.

- **ReAct basic.** Navigation and raw span-editing actions (`list_headings`, `search`, `search_lines`, `view`, `view_lines`, `replace_span`, `insert_span`, `delete_span`, `cut`, `paste`, `finish`) with an 8-step budget. Edits operate on character offsets directly.

Both variants use `gpt-4.1` and `grok-4-1-fast-reasoning`. The ReAct+`apply_patch` rows used in Table 2 cover 10 document IDs with five repetitions per document for each model/condition; the appended expansion uses a 120 s per-request timeout. Appendix E reports per-condition detail tables for both ReAct variants, including retained diagnostic rows outside the main comparison.

**Per-task pattern.** The ReAct+`apply_patch` variant approaches Functional Safety on targeted paragraph editing (98% success, 3% side effects) and remains competitive on paragraph consolidation (66–100% success

depending on document size and model, $\leq 6\%$ side effects). However, it has lower observed success and higher side-effect rates on structural tasks: code visibility refactors (0–2% success, 14–52% side effects), section relocation (0–46% success, 42–98% side effects), and retained section-reordering diagnostics (0–20% success). On LaTeX micro-edits, the apply_patch variant achieves 73% aggregate success with 27% aggregate side effects: simple math-symbol fixes succeed reliably (98–100% with near-zero SE), but multi-location edits (cross-references requiring coordinated `\label`/`\ref` changes: 24–98% success; citation keys: 24–90% success) show high variance across models and produce frequent intra-target side effects where the agent edits the wrong location within the target chapter. The basic variant performs worse in most evaluated conditions, with higher side-effect rates from raw-offset corruption—and the gap is strongly model-dependent: `gpt-4.1` achieves near-0% success on basic tools while `grok` reaches 60–100% on many tasks, suggesting that extended reasoning can compensate for raw-offset arithmetic. The `apply_patch` tool narrows this gap via context matching, converging on easy tasks but leaving a residual divergence on LaTeX multi-location edits where precise document navigation still matters (cross-references: `grok` 88–98% vs. `gpt-4.1` 24–26%; citations: 88–90% vs. 24–38%). This gradient suggests that patch-based tool editing remains vulnerable on evaluated tasks requiring coordinated changes or structural reorganization.

## 5.6 Cross-Task Synthesis

To unify the empirical findings across domains, we include a narrative synthesis of success rates, latency, and failure modes:

Functional Safety decomposes edits into typed plans over explicit hierarchy units and executes accepted plans with deterministic locality checks. Flat baselines rewrite larger spans directly, while ReAct tool agents apply edits through an observe–think–act loop with span-editing or patch-based tools (Section 5.5).

Table 2 summarizes strict success and strict side-effect rates across the six main remote task families: targeted paragraph editing, contextual insert, code visibility refactor, long-form relocation, paragraph consolidation, and LaTeX micro-edits. The flat/FS rows use `gpt-4.1` and `grok-4-1-fast-reasoning` over 10 document IDs with five repetitions per document for the reported single-document conditions. ReAct sample sizes are reported explicitly in the table.

**Main trend.** The flat baseline has near-zero strict success because full-span regeneration often changes untargeted regions, even when the requested content is present under whitespace-normalized scoring. Re-Act+`apply_patch` remains competitive on localized paragraph edits and paragraph consolidation, but degrades on structural moves, code reordering, and multi-location LaTeX edits. Functional Safety reduces observed strict side effects to zero in the reported GPT/Grok rows. Its remaining failures are task-completion failures from planner or worker conformance, not executor-side locality failures.

**Prose edits and model breadth.** For targeted paragraph editing, Functional Safety reaches 85/100 strict success across GPT/Grok with 0/100 strict side effects. For contextual insert, it reaches 76/100 strict success with 0/100 strict side effects. The main remote comparison is restricted to model/provider configurations that completed the complete main remote suite under the fixed evaluation harness.

**Structural and markup edits.** For code visibility refactors, long-form section relocation, paragraph consolidation, and LaTeX micro-edits, the reported GPT/Grok Functional Safety rows have 0 observed strict side effects in every reported condition. `gpt-4.1` reaches 50/50 strict success in each reported row, while `grok-4-1-fast-reasoning` reaches 35/50 in the same rows. This makes the empirical story more precise: the executor/locality mechanism is robust in these runs, but planner reliability still depends on the model and task.

**Local/open models.** The local/open-model rows cover three smaller tasks using two run budgets: 15 runs per task for `llama3:8b`, `qwen2.5:14b`, and `gemma3:27b`, and 50 runs per task for `qwen3.5:27b`. All four evaluated local models solve these tasks under FS with zero observed strict side effects: `llama3:8b`, `qwen2.5:14b`, and `gemma3:27b` each reach 45/45 strict success, while `qwen3.5:27b` in the non-reasoning Ollama setting reaches 150/150. The flat baseline has zero strict success in these rows because full-span rewrites change untargeted regions. Its whitespace-normalized success is higher for some local models (`qwen2.5:14b`: 15/45, `gemma3:27b`: 30/45, `qwen3.5:27b`: 100/150), showing that some requested edits survive after format-

ting differences are ignored. The reasoning-mode sensitivity check is worse for FS task completion because it increases latency and structured-output failures; those failures still do not introduce side effects.

**Tool-agent pattern.** The ReAct+`apply_patch` agent approaches Functional Safety on some localized edits, but the gap widens when the task requires structural reorganization or coordinated edits across multiple markup locations. This supports the paper's central distinction: context-matching patches can be effective local edit tools, but they do not provide a document-level hierarchy, typed structural operators, or deterministic locality enforcement.

**Multi-file refactors.** For Python and LATEX multi-file reconstruction, we report identity metrics, import and signature deltas, tests when available, and latency. Exact byte identity is hardest in open-layout and imports-top variants, but test-oriented behavior preservation remains high in several settings. The multi-file tables should therefore be read jointly because identity, deltas, and tests capture different aspects of refactor quality.

## 6 Related Work

Research at the intersection of hierarchical modeling, structured editing, and LLM safety spans several partially overlapping communities. We focus on six strands of work that illuminate different facets of the problem: long-context and hierarchical Transformers, structure-aware code representations, planner–executor and tool-using LLMs, constrained generation and document understanding, safety and guardrail frameworks, and efficient long-sequence models. Our aim is not to survey each area exhaustively, but to clarify why existing approaches do not provide the symbolic, functionally constrained editing architecture developed in this paper.

### 6.1 Long-Context and Hierarchical Transformers

A natural response to the structural limitations of LLMs is to extend context length or incorporate multi-scale attention. Models such as Longformer and BigBird introduce sparse attention patterns that allow tens of thousands of tokens to be processed efficiently while maintaining strong document-level performance Beltagy et al. (2020); Zaheer et al. (2020). Hierarchical attention variants further segment inputs into sentences or paragraphs and perform cross-segment attention to capture multi-level structure Pappagari et al. (2019); Chalkidis et al. (2022); He et al. (2024). These architectures can be viewed as neural analogues to the two-level representation discussed in our introduction: one mechanism attends within segments, another across them.

However, these models treat hierarchy as a latent internal structure rather than an explicit symbolic object. They do not explicitly manipulate named units—e.g., "the third paragraph in section 2" or "lemma 4"—which makes it difficult to provide formal assurances that edits remain localized or that protected regions' content remains unchanged. By contrast, the approach in this paper exposes hierarchy explicitly and can route modifications through a symbolic plan interpreted by an executor that enforces structural invariants. Our method is therefore complementary: long-context or hierarchical Transformers may serve as powerful planners, and embedding them in a structure-aware execution framework can provide editing assurances in this setting.

### 6.2 Structure-Aware Code Models and Tree Encoders

A second line of work incorporates syntax and control-flow information to improve code understanding. Models such as CodeBERT and GraphCodeBERT augment token sequences with data-flow or semantic graphs for code search and summarization Feng et al. (2020); Guo et al. (2021). Tree-based models such as TreeBERT and AST-Transformer go further by encoding abstract syntax trees directly Jiang et al. (2021); Tang et al. (2021). These approaches demonstrate that explicit structure can significantly improve robustness on program-analysis tasks.

Yet even in tree-aware models, editing remains predominantly autoregressive: the model regenerates AST fragments or textual spans, and there is no formal mechanism preventing unintended changes outside the

intended region. Moving a function or reordering a block may inadvertently modify formatting, comments, or semantically adjacent code. Our work differs in kind: edits are expressed as symbolic operators over hierarchical nodes, and application is carried out by a deterministic executor that isolates stochasticity to planner-selected regions, preserves byte-for-byte payload outside each step's payload footprint, and confines structural changes to each step's structural footprint under the stated assumptions. This yields stronger refactoring assurances in this setting than are typically available in prior code-model architectures.

### 6.3 Planner–Executor Architectures and Tool-Using LLMs

Planner–executor decompositions have become central to extending LLM capabilities. ReAct couples chain-of-thought reasoning with tool invocation Yao et al. (2023), while Toolformer shows that LLMs can self-supervise their own use of external APIs Schick et al. (2023). Recent software-engineering agents such as SWE-agent and OpenHands extend this paradigm with agent-computer interfaces, command execution, file editing, and benchmarked repository-level workflows Yang et al. (2024); Wang et al. (2025b).

These systems share a core intuition with our work: language models should propose actions, and a separate component should execute them. However, tool invocations in existing frameworks typically operate at the level of free-form text manipulation or broad task descriptions ("edit this paragraph," "summarize this section"). They do not act over discrete hierarchical units, nor do they provide symbolic edit operators with formally specified invariants. In contrast, our methodology treats the planner's outputs as typed edit plans over a structured hierarchy and constrains worker modules—potentially stochastic themselves— to operate within the regions selected by the planner. A deterministic executor then verifies structure, enforces locality, and applies the permissible changes specified by the plan. We implement ReAct-style baselines—including a context-matching `apply_patch` variant—as concrete comparators (Section 5.5). In our experiments, ReAct+`apply_patch` approaches Functional Safety on simple localized edits but performs worse on evaluated structural tasks (reordering, relocation), where explicit hierarchy appears most useful.

### 6.4 Constrained Generation and Structured Document Understanding

Constrained decoding and guided-generation systems enforce syntactic constraints such as regular expressions, JSON schemas, or context-free grammars during token generation Willard & Louf (2023); Dong et al. (2025b). These methods are useful for making planner outputs parseable, and they are complementary to the plan-validation layer in Functional Safety. However, syntactically valid output is not the same as a locality guarantee: a valid JSON span edit can still select the wrong offset or overbroad region unless execution is tied to explicit document units and footprint checks.

Structured document-understanding models similarly address an adjacent dependency. Layout-aware models such as DocLLM reason over visual and textual layout to recover document semantics from complex inputs Wang et al. (2024). Such systems may improve hierarchy extraction, especially for visually rich or weakly structured documents. They do not by themselves specify how edits should be applied once a hierarchy is obtained. Functional Safety treats extraction as an explicit upstream contract and focuses on the downstream execution semantics for typed edits over the extracted hierarchy.

### 6.5 Safety, Guardrails, and Model Editing

A growing literature examines safety vulnerabilities in unconstrained LLM outputs and model updates. Surveys highlight risks associated with jailbreaks, prompt injection, and training-set poisoning, motivating stronger guardrails and programmable layers between users and foundation models Dong et al. (2025a); Zhang et al. (2025); Liu et al. (2025). Parallel work in knowledge editing studies techniques for modifying specific model facts or behaviors, but often uncovers unintended global side effects on generalization and downstream tasks Wang et al. (2025a); Hsueh et al. (2024); Youssef et al. (2025).

These approaches generally treat safety as a property of either the model's internal parameters or a wrapper around its input/output channels. They do not typically provide fine-grained structural assurances for document or code transformations. Our approach is orthogonal: by reifying structure into explicit hierarchies and restricting edits to a deterministic executor that applies a typed edit algebra, we obtain functional-style

invariants—non-interference, structural preservation, and bounded stochasticity—for validated plans over correctly extracted hierarchies. This shifts a portion of safety reasoning from opaque parameter space into a discrete, auditable space of symbolic operations.

### 6.6 Relation to Efficient Long-Sequence Models

A rich body of work on *efficient long-sequence models* explores sparse, local, or hierarchical attention mechanisms and state-space alternatives to reduce computational cost for long sequences Beltagy et al. (2020); Zaheer et al. (2020); Kitaev et al. (2020); Wang et al. (2020); Gu & Dao (2023). These models compress attention patterns or introduce state-space abstractions to effectively skim long inputs while attending selectively to relevant regions.

Our methodology is complementary. Rather than modifying attention patterns inside the model, we construct an explicit symbolic hierarchy outside the model and can restrict the planner and workers to operate on trimmed or summarized node views. This constitutes an application-level analogue of sparse attention: the model need not process entire documents unless the task calls for it, and execution is hardened by structural constraints.

### 6.7 Cognitive and Functional Programming Perspectives

Finally, our work draws on two bodies of ideas rarely combined in LLM research: hierarchical models of human cognition and the design principles of functional programming. Cognitive theories emphasize chunking, multi-level representation, and compositional reasoning when humans manipulate complex artifacts such as proofs or codebases. Functional programming emphasizes purity, composability, and explicit control of side effects.

The architecture developed in this paper synthesizes these perspectives by (i) representing documents and programs as explicit hierarchies of semantic units and (ii) enforcing that edits occur through a deterministic executor that isolates model stochasticity—both from the planner and from worker modules—within carefully bounded regions. To our knowledge, no existing LLM framework integrates symbolic hierarchical representations with a formally specified, structure-constrained execution layer. Filling this gap is the central motivation of the present work.

## 7 Discussion

The proposed architecture unifies insights from hierarchical cognition, functional programming, and contemporary LLM design into a structured framework for controlled editing and reasoning. By introducing explicit symbolic units and a deterministic executor, the system aims to address limitations of token-level generation—particularly structural drift, collateral modifications outside the intended scope, and the lack of formal assurances about edit locality in many settings.

At a conceptual level, the architecture bridges human-like hierarchical reasoning with machine representations. Humans naturally operate over multi-level abstractions—paragraphs, sections, arguments, lemmas—rather than individual symbols. By mirroring this structure explicitly, the system exposes units over which models can plan and tools can act.

From a systems perspective, the separation between planning and execution introduces a principled notion of constrained editing. The planning stage performs high-level symbolic reasoning, while the executor enforces safety properties through purity and determinism in this formulation. This mirrors design principles in functional programming, where side effects are controlled by isolating mutation behind explicit interfaces. Such modularity can improve reliability in settings where the hierarchy and plan contracts are satisfied, and it facilitates auditing, verification, and interpretability.

The empirical comparison with ReAct-style tool-using agents clarifies the limitations of that architecture compared to an approach which uses explicit hierarchies. Although ReAct can do quite well in many tasks, and even has relatively low side effect rates in some tasks, the side effect rates are still strictly non-zero. Up

to this point most research has focused on improving outcomes directly through improving the intelligence of language models, whereas the FS approach suggests architectural changes may be useful in producing stronger guarantees. Although a more intelligent model may be less likely to make mistakes, there can still be value in using methodologies where there are stronger guarantees.

## 7.1 Failure Modes and Challenges

Functional Safety reduces some classes of unintended edits by replacing free-form whole-document generation with typed plans and deterministic execution, but it does not remove every failure mode. Failures can enter at three interfaces.

**Extraction.** Although hierarchy is a natural feature of many document types, including computer code, there could arise situations where documents contain issues which make correct hierarchy extraction difficult. For example, we experimented with documents obtained through OCR which introduces artifacts into a document and cannot perfectly recover the structure.

**Planning.** The router can emit malformed JSON, or can create a plan with the wrong scope. These challenges are more likely as the size of documents increases, including the complexity of the hierarchy. Tracking and manipulating complex hierarchies can produce long or complex JSON on which language models struggle more. Although the malformed JSON is more readily detectable and the operation can be rejected, situations where the JSON is correctly specified but the scope or edit intent is incorrect are much more difficult to detect.

**Worker execution.** Worker modules can violate local output contracts even when the plan is correct. For example, a worker may emit multiple paragraphs for a single-paragraph insert, or return unparsable markup. Guardrails and validation can fail closed in many of these cases, but fail-closed behavior trades side-effect prevention for lower task success.

**Fail-closed behavior and model dependence.** Validation-checkable planner errors and worker contract violations are captured by the executor as *fail-closed* outcomes rather than as side effects. Semantic planner errors can in principle produce side effects when the wrong target lies outside the task-defined region, but across all six remote task families and both evaluated models the observed FS side-effect rate in Table 2 is zero, whereas ReAct+`apply_patch` has non-zero side effect rates. The associated task-success behavior is model-dependent: `gpt-4.1` produces valid FS plans on essentially every trial, reaching 100% success on five of six task families (82% on Contextual Insert), whereas `grok-4-1-fast-reasoning` sits at 70% across all six families—the same ceiling its flat-baseline rewrites hit under WS on the short prose tasks. Among the evaluated remote models, only `gpt-4.1` under FS achieves both high task success and zero observed side effects across all six families; `grok-4-1-fast-reasoning` under FS sacrifices roughly 30% of attempts to non-success outcomes (fail-closed rejections, or accepted edits whose footprint stayed within the task region but did not satisfy the task oracle) rather than to corruption outside the target. This is the regime FS targets: bounding the consequences of imperfect planners and workers, not improving their underlying content reliability.

**Overhead.** The method also has a cost-benefit boundary. Maintaining a hierarchy, carrying node identifiers and previews, producing JSON plans, and validating footprints all add latency and token overhead. For short documents or simple single-span edits, a context-matching patch may be adequate and cheaper. The advantages of Functional Safety are clearest when edits require structural moves, multi-location coordination, long-document locality, or high confidence that untouched regions remain unchanged.

## 7.2 Implications and Open Questions

The architecture also has implications for AI safety. Existing methods for constraining model behavior often operate post hoc—inspecting or evaluating outputs after generation. In contrast, the symbolic executor enforces constraints during the transformation process itself, so that safety properties such as locality, payload

non-interference, and structural soundness hold by construction in the formal model rather than by inspection. When a well-formed plan is executed, payload side effects outside each step's payload footprint are shown to be absent in the formal model under the stated assumptions; when a malformed plan is submitted, the executor rejects it and performs no modification. This shift from reactive to proactive safety design provides a conditional formal component that could be useful in high-stakes workflows only with additional validation, deployment safeguards, and clear communication of the assumptions.

Finally, the framework raises new research questions. How can hierarchical representations be induced automatically and optimally? How expressive should the edit algebra be across domains? Can symbolic plans be optimized or verified beyond local invariants? And how does this architecture interact with emerging long-context and retrieval-based systems? Exploring these questions may lead to LLMs that not only generate text but also manipulate structured knowledge with the precision and reliability found in human reasoning.

## 8 Limitations

Despite its advantages, the proposed architecture also has several limitations and open challenges. First, it assumes access to a reliable mechanism for constructing hierarchical representations. Many structures—such as sections, paragraphs, or code functions—can be identified through heuristics or existing parsers, but more abstract or noisy domains may call for sophisticated analysis pipelines whose correctness becomes an additional dependency.

Second, the architecture introduces representational overhead. Deterministic hierarchy extraction is usually inexpensive, but the extracted structure must be serialized into prompts, plans, and validation artifacts. For large or highly fragmented documents, the JSON representation of node identifiers, paths, summaries, and operator arguments can become large relative to the edit itself. This increases token cost and latency, and can make planning less reliable when the model must reason over or emit bulky structured objects. Practical deployments therefore need compact hierarchy summaries, retrieval over relevant subtrees, and domain-specific serialization formats rather than naively passing the full structure to the planner.

Third, the main remaining failure modes are planner errors and worker errors. The planner may choose the wrong operation, target the wrong node, select an overly broad scope, or omit a required step. The worker may produce local replacement text that does not satisfy the requested edit or violates a narrow output contract. The executor can reject malformed plans and unacceptable worker outputs, and it can enforce locality for accepted operations, but it does not infer the user's intent or repair a conceptually wrong plan. Functional safety is therefore conditional: for a correctly extracted hierarchy, a valid plan, and accepted local worker outputs, execution is locality-bounded; failures before that point remain planning or local-generation failures.

Finally, the architecture does not eliminate the need for broader safety measures. It is best viewed as complementing, rather than replacing, work on interpretability, robust evaluation, and adversarial testing. Full integration into safety-critical systems would still call for additional safeguards beyond the structural assurances discussed here.

### 8.1 Future Work

Several promising directions emerge from this framework. First, implementing the full architecture in small- or medium-scale LLMs would allow empirical evaluation of the tradeoffs between symbolic planning and autoregressive generation, particularly for long-context editing tasks. Second, automating hierarchy induction remains an open challenge: many domains provide natural structural boundaries, but others may call for learned or hybrid parsing mechanisms capable of identifying meaningful units without manual heuristics.

Third, enriching the edit algebra with domain-specific operations could further expand the system's applicability while preserving safety assurances. We already demonstrate semantic reorganization, cross-referencing, and multi-document transformations; future work is to generalize these capabilities into broader operator families and richer domain-specific semantics. Fourth, integrating the architecture with retrieval-based or

memory-augmented models may enable more sophisticated reasoning over distributed or external knowledge sources.

In our current implementation, the router receives fixed-size previews of each hierarchy node. An appealing extension is to use an adaptive scanning scheme in which the model reads sections incrementally and is periodically asked whether it has seen enough to make a structural decision. If the answer is yes, scanning stops early; otherwise, additional tokens are streamed. This could further reduce context length and latency, especially for long or repetitive sections.

## 9 Conclusion

This work describes a Functional Safety approach for large language models, separating analysis, planning, and execution into distinct components. We prove a Deterministic Safety theorem (Theorem 5) showing that, under the formal model and stated assumptions, correctly extracted hierarchies and valid symbolic plans are payload-side-effect-free outside each step's payload footprint (and thus outside the plan-level union). By introducing explicit symbolic units and delegating structural edits to a deterministic executor, the approach addresses limitations of flat token-level generation with formal properties and empirical evidence of side-effect reduction, structural consistency, and execution invariants.

The multi-stage pipeline leverages the strengths of modern LLMs while constraining their weaknesses. The host-built hierarchy provides the data needed for symbolic planning; the router emits typed operations; and the executor enforces invariants as it moves or edits sections. Empirically, Functional Safety improves observed strict success rates and lowers observed side-effect rates in the evaluated single-document conditions, especially on structural and multi-location edits where the representative ReAct-style patch baseline degrades. This division can improve reliability in tasks such as document editing, code refactoring, and structured reasoning, while leaving hierarchy extraction, planning accuracy, and worker-output conformance as explicit boundary conditions.

More broadly, the procedure connects insights from cognitive science and functional programming with ongoing efforts in AI safety, interpretability, and agentic LLMs. By shifting from implicit to explicit hierarchy, and from direct generation to verifiable transformation, it points toward systems that manipulate structured knowledge with greater reliability and control.

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

## A   Procedure Details

This appendix reports only the core orchestration procedures used by the method: (1) typed plan synthesis/validation and (2) deterministic execution with worker-output guardrails and invariant checks. Auxiliary engineering variants (e.g., grouped planning and open-ended layout selection) are described in prose in Section 3.

---

**Procedure 1** Typed Plan Synthesis and Validation

---

**Require:** Hierarchy handle $H$, user instruction $q$, optional task metadata $m$
**Ensure:** Well-typed symbolic plan $P$ or error
  1: $(S, \text{prefix}, \text{suffix}) \leftarrow \text{EXTRACTSECTIONINFOS}(H)$
  2: Optionally run lightweight locator to produce disambiguation hints $L$
  3: Optionally compute sparse content cues $H_s$ from $q$ and section summaries
  4: **if** $H_s$ is marked ambiguous and ambiguity guard is enabled **then**
  5:   **return** ERROR(AmbiguousTarget)
  6: **end if**
  7: Construct router prompt with allowed operators $\mathcal{O}$, summaries, $L$, optional $H_s$, and instruction $q$
  8: $(\text{raw\_plan}, \text{meta}) \leftarrow \text{LLMCOMPLETE}(rllm, \text{prompt})$
  9: Parse raw\_plan into candidate plan object
 10: **if** parse fails **then**
 11:   **return** ERROR(InvalidPlanEncoding)
 12: **end if**
 13: **if** candidate delegates pure permutation **then**
 14:   $\pi \leftarrow \text{EXTRACTPERMUTATION}(\text{candidate})$
 15:   **if** $\pi$ is not a permutation of $S$ **then**
 16:     **return** ERROR(InvalidPermutation)
 17:   **end if**
 18:   **return** $(\text{MOVE}(\pi), \text{meta})$
 19: **end if**
 20: $P_{\text{raw}} \leftarrow$ ordered operations extracted from candidate
 21: **for** each operation $o \in P_{\text{raw}}$ **do**
 22:   **if** $o$ is MOVE($\pi$) and $\pi$ is not a permutation of $S$ **then**
 23:     **return** ERROR(InvalidPermutation)
 24:   **else if** $o$ targets nonexistent node IDs or invalid positions **then**
 25:     **return** ERROR(InvalidTarget)
 26:   **else if** $o$ contains invalid label/type fields **then**
 27:     **return** ERROR(InvalidOperator)
 28:   **end if**
 29: **end for**
 30: $P \leftarrow \text{NORMALIZEPLAN}(P_{\text{raw}})$
 31: **return** $(P, \text{meta})$

---

---

**Procedure 2** Deterministic Execution with Guardrails and Invariants
___

**Require:** Hierarchy handle $H$, well-typed plan $P$, optional intended structural footprint $T$
**Ensure:** Updated document text $X'$ or error
1: $X' \leftarrow H.\text{doc\_text}$
2: **for** each operation $o$ in $P$ **do**
3:    Compute per-step structural footprint $F_{\text{struct}}(o)$ and payload footprint $F_{\text{payload}}(o)$
4:    **if** $T$ is provided and $F_{\text{struct}}(o) \nsubseteq T$ **then**
5:       **return** ERROR(`InvalidFootprint`)
6:    **end if**
7:    **if** $o$ is MOVE **then**
8:       Apply deterministic reordering/reparenting update
9:    **else if** $o$ is EDIT **then**
10:       Extract target content (crop if specified)
11:       **if** replacement content is not provided in plan **then**
12:          Call worker with scoped context
13:          Normalize wrapper artifacts (fences/labels/quotes)
14:          **if** guardrails reject output and retry budget remains **then**
15:             Retry worker call
16:          **end if**
17:          **if** guardrails still reject output **then**
18:             **return** ERROR(`WorkerOutputInvalid`)
19:          **end if**
20:       **end if**
21:       Splice accepted content into target span
22:       Re-extract affected subtree when structural refresh is required
23:    **else if** $o$ is INSERT **then**
24:       Materialize inserted content (worker call only if content omitted)
25:       Insert new node at validated position
26:    **else if** $o$ is DELETE **then**
27:       Remove node and descendants
28:    **else if** $o$ is ANNOTATE **then**
29:       Update node label/metadata without changing payload
30:    **else**
31:       **return** ERROR(`UnknownOperation`)
32:    **end if**
33:    **if** payload outside $F_{\text{payload}}(o)$ changed **then**
34:       **return** ERROR(`PayloadViolation`)
35:    **end if**
36:    **if** structural invariants fail (nesting/acyclicity/non-overlap) **then**
37:       **return** ERROR(`InvariantViolation`)
38:    **end if**
39: **end for**
40: **return** $X'$

---

**Execution note.**   The executor operates over normalized typed operations and fails closed on invalid plans, invalid worker outputs (after bounded retries), or invariant violations. This is the mechanism behind the step-wise non-interference claims in Section 4.

## B   Local Model Experiment Details

This appendix provides detailed experimental results for local models on the smaller tasks. These experiments illustrate that Functional Safety can make local models more usable editors on these tasks by confining edits to the intended regions. Local results are reported in Tables 9, 10, and 11. The first three local models

use 15 runs per condition; `qwen3.5:27b` uses 50 runs over 10 document IDs per condition. We report these rows descriptively rather than as a model-scaling analysis; the evaluated local backends differ in architecture, serving stack, context handling, and structured-output behavior.

## B.1 `qwen3.5:27b` Reasoning-Mode Sensitivity

We also evaluated `qwen3.5:27b` under Ollama's reasoning-mode interface on the same three smaller tasks. The main local-model comparison uses the non-reasoning setting, because it is the cleaner structured-output condition for this model. A first reasoning-mode pilot with the default completion budget frequently failed to produce parseable JSON; the sensitivity run in Table 8 therefore uses a 4096-token completion cap. Results are reported with whitespace-normalized metrics because the local flat baseline often changes formatting.

Table 8: `qwen3.5:27b` reasoning-mode sensitivity (50 runs over 10 document IDs per condition). Reasoning mode increases latency and structured-output failures; Functional Safety still has zero observed whitespace-normalized side effects.

| Task | Mode | Flat Succ. (WS) | Flat SE (WS) | FS Succ. (WS) | FS SE (WS) | FS Time (s) | FS non-ok status |
|---|---|---|---|---|---|---|---|
| Targeted Paragraph Editing | think=false | 50/50 | 0/50 | 50/50 | 0/50 | 2.5 | 0/50 |
| Targeted Paragraph Editing | think=true | 50/50 | 0/50 | 45/50 | 0/50 | 46.5 | worker_error: 5/50 |
| Contextual Insert | think=false | 50/50 | 0/50 | 50/50 | 0/50 | 3.8 | 0/50 |
| Contextual Insert | think=true | 36/50 | 14/50 | 21/50 | 0/50 | 91.0 | worker_error: 28/50 |
| Code Visibility Refactor | think=false | 0/50 | 50/50 | 50/50 | 0/50 | 4.1 | 0/50 |
| Code Visibility Refactor | think=true | 0/50 | 50/50 | 25/50 | 0/50 | 91.4 | parse_error: 25/50 |

The reasoning-mode failures are not side-effect failures. They are planner or worker conformance failures that the executor treats as fail-closed outcomes. In the prose insertion task, the router often emits a valid paragraph-level INSERT plan, but the worker returns an empty or rejected paragraph, so no edit is applied. In the code visibility task, reasoning-mode calls frequently consume the full completion budget before exposing a parseable JSON plan, producing `parse_error` outcomes. This sensitivity check reinforces the failure-mode discussion in Section 7.1: Functional Safety bounds the effects of accepted operations, but it does not guarantee that a stochastic planner or worker will always produce a usable structured output.

## B.2 Targeted Paragraph Editing

Table 9: Targeted Paragraph Editing (local models). Functional Safety preserves zero observed side effects; task completion varies by model.

| Method | Success Rate (WS) | Success Rate (Strict) | Side-Effect Rate (WS) | Side-Effect Rate (Strict) | Δ Strict Success | p (Strict Succ) | p (Strict SE) | Time (s) |
|---|---|---|---|---|---|---|---|---|
| llama3:8b (Flat) | 0.00 [0.00, 0.20] | 0.00 [0.00, 0.20] | 1.00 [0.80, 1.00] | 1.00 [0.80, 1.00] | – | – | – | 1.96 |
| llama3:8b (Functional Safety) | 1.00 [0.80, 1.00] | 1.00 [0.80, 1.00] | 0.00 [0.00, 0.20] | 0.00 [0.00, 0.20] | +1.00 [+0.71, +1.00] | $< 10^{-3}$ | $< 10^{-3}$ | 0.63 |
| qwen2.5:14b (Flat) | 1.00 [0.80, 1.00] | 0.00 [0.00, 0.20] | 0.00 [0.00, 0.20] | 1.00 [0.80, 1.00] | – | – | – | 3.46 |
| qwen2.5:14b (Functional Safety) | 1.00 [0.80, 1.00] | 1.00 [0.80, 1.00] | 0.00 [0.00, 0.20] | 0.00 [0.00, 0.20] | +1.00 [+0.71, +1.00] | $< 10^{-3}$ | $< 10^{-3}$ | 0.89 |
| gemma3:27b (Flat) | 1.00 [0.80, 1.00] | 0.00 [0.00, 0.20] | 0.00 [0.00, 0.20] | 1.00 [0.80, 1.00] | – | – | – | 6.40 |
| gemma3:27b (Functional Safety) | 1.00 [0.80, 1.00] | 1.00 [0.80, 1.00] | 0.00 [0.00, 0.20] | 0.00 [0.00, 0.20] | +1.00 [+0.71, +1.00] | $< 10^{-3}$ | $< 10^{-3}$ | 1.98 |
| qwen3.5:27b (Flat) | 1.00 [0.93, 1.00] | 0.00 [0.00, 0.07] | 0.00 [0.00, 0.07] | 1.00 [0.93, 1.00] | – | – | – | 7.64 |
| qwen3.5:27b (Functional Safety) | 1.00 [0.93, 1.00] | 1.00 [0.93, 1.00] | 0.00 [0.00, 0.07] | 0.00 [0.00, 0.07] | +1.00 [+0.86, +1.00] | $< 10^{-3}$ | $< 10^{-3}$ | 2.48 |

## B.3 Contextual Insert

Table 10: Contextual Insert (local models). Functional Safety preserves zero observed side effects; task completion varies by model.

| Method | Success Rate (WS) | Success Rate (Strict) | Side-Effect Rate (WS) | Side-Effect Rate (Strict) | Δ Strict Success | p (Strict Succ) | p (Strict SE) | Time (s) |
|---|---|---|---|---|---|---|---|---|
| llama3:8b (Flat) | 0.00 [0.00, 0.20] | 0.00 [0.00, 0.20] | 1.00 [0.80, 1.00] | 1.00 [0.80, 1.00] | – | – | – | 1.26 |
| llama3:8b (Functional Safety) | 1.00 [0.80, 1.00] | 1.00 [0.80, 1.00] | 0.00 [0.00, 0.20] | 0.00 [0.00, 0.20] | +1.00 [+0.71, +1.00] | $< 10^{-3}$ | $< 10^{-3}$ | 0.74 |
| qwen2.5:14b (Flat) | 0.00 [0.00, 0.20] | 0.00 [0.00, 0.20] | 1.00 [0.80, 1.00] | 1.00 [0.80, 1.00] | – | – | – | 3.30 |
| qwen2.5:14b (Functional Safety) | 1.00 [0.80, 1.00] | 1.00 [0.80, 1.00] | 0.00 [0.00, 0.20] | 0.00 [0.00, 0.20] | +1.00 [+0.71, +1.00] | $< 10^{-3}$ | $< 10^{-3}$ | 1.03 |
| gemma3:27b (Flat) | 1.00 [0.80, 1.00] | 0.00 [0.00, 0.20] | 0.00 [0.00, 0.20] | 1.00 [0.80, 1.00] | – | – | – | 6.32 |
| gemma3:27b (Functional Safety) | 1.00 [0.80, 1.00] | 1.00 [0.80, 1.00] | 0.00 [0.00, 0.20] | 0.00 [0.00, 0.20] | +1.00 [+0.71, +1.00] | $< 10^{-3}$ | $< 10^{-3}$ | 3.15 |
| qwen3.5:27b (Flat) | 1.00 [0.93, 1.00] | 0.00 [0.00, 0.07] | 0.00 [0.00, 0.07] | 1.00 [0.93, 1.00] | – | – | – | 7.28 |
| qwen3.5:27b (Functional Safety) | 1.00 [0.93, 1.00] | 1.00 [0.93, 1.00] | 0.00 [0.00, 0.07] | 0.00 [0.00, 0.07] | +1.00 [+0.86, +1.00] | $< 10^{-3}$ | $< 10^{-3}$ | 3.78 |

## B.4 Code Visibility Refactor

Table 11: Code Visibility Refactor (local models, small classes). Functional Safety preserves zero observed side effects; task completion varies by model.

| Method | Success Rate (WS) | Success Rate (Strict) | Side-Effect Rate (WS) | Side-Effect Rate (Strict) | Δ Strict Success | p (Strict Succ) | p (Strict SE) | Time (s) |
|---|---|---|---|---|---|---|---|---|
| llama3:8b (Flat) | 0.00 [0.00, 0.20] | 0.00 [0.00, 0.20] | 1.00 [0.80, 1.00] | 1.00 [0.80, 1.00] | – | – | – | 1.58 |
| llama3:8b (Functional Safety) | 1.00 [0.80, 1.00] | 1.00 [0.80, 1.00] | 0.00 [0.00, 0.20] | 0.00 [0.00, 0.20] | +1.00 [+0.71, +1.00] | $< 10^{-3}$ | $< 10^{-3}$ | 1.10 |
| qwen2.5:14b (Flat) | 0.00 [0.00, 0.20] | 0.00 [0.00, 0.20] | 1.00 [0.80, 1.00] | 1.00 [0.80, 1.00] | – | – | – | 3.20 |
| qwen2.5:14b (Functional Safety) | 1.00 [0.80, 1.00] | 1.00 [0.80, 1.00] | 0.00 [0.00, 0.20] | 0.00 [0.00, 0.20] | +1.00 [+0.71, +1.00] | $< 10^{-3}$ | $< 10^{-3}$ | 1.65 |
| gemma3:27b (Flat) | 0.00 [0.00, 0.20] | 0.00 [0.00, 0.20] | 1.00 [0.80, 1.00] | 1.00 [0.80, 1.00] | – | – | – | 7.28 |
| gemma3:27b (Functional Safety) | 1.00 [0.80, 1.00] | 1.00 [0.80, 1.00] | 0.00 [0.00, 0.20] | 0.00 [0.00, 0.20] | +1.00 [+0.71, +1.00] | $< 10^{-3}$ | $< 10^{-3}$ | 3.73 |
| qwen3.5:27b (Flat) | 0.00 [0.00, 0.07] | 0.00 [0.00, 0.07] | 1.00 [0.93, 1.00] | 1.00 [0.93, 1.00] | – | – | – | 7.68 |
| qwen3.5:27b (Functional Safety) | 1.00 [0.93, 1.00] | 1.00 [0.93, 1.00] | 0.00 [0.00, 0.07] | 0.00 [0.00, 0.07] | +1.00 [+0.86, +1.00] | $< 10^{-3}$ | $< 10^{-3}$ | 4.06 |

# C Hierarchy Extraction

This appendix reports only the hierarchy-extraction components that are central to the empirical claims in the paper. For structured formats (Markdown, LaTeX, Python, and related code formats), we use deterministic parsers and syntax-driven span assignment. For weakly structured documents, we use the two-pass LLM-assisted extraction procedure below while keeping paragraph spans deterministic.

Whitespace policies for lossless rendering and strict side-effect accounting are described in Appendix D.

## C.1 LLM extraction diagnostics on structured documents

We compare LLM-extracted hierarchies against deterministic parsers on a small structured fixture set (Markdown and LaTeX, four documents) with 15 repetitions per model. Table 12 reports success counts, span F1 at IoU 0.9 for top-level sections and paragraphs, best-IoU averages, and mean latency. Across models, LLM extraction is 4–5 orders of magnitude slower than host parsing. `grok-4-1-fast-reasoning` yields the strongest section alignment, while both models remain well below deterministic parsers on span F1. These results underscore the latency cost and span fragility of LLM-first hierarchy extraction on structured inputs.

Table 12: LLM hierarchy extraction vs. deterministic parsing on structured fixtures (15 repetitions per model).

| Model | Format | Docs | Sec F1@0.9 | Sec IoU | Para F1@0.9 | Para IoU | LLM mean (s) | Host mean (s) |
|---|---|---|---|---|---|---|---|---|
| gpt-4.1 | overall | 60/60 | 0.21 | 0.69 | 0.38 | 0.67 | 2.36 | 0.00012 |
| gpt-4.1 | latex | 30/30 | 0.50 | 0.89 | 0.00 | 0.40 | 2.60 | 0.00012 |
| gpt-4.1 | markdown | 30/30 | 0.04 | 0.56 | 0.67 | 1.00 | 2.12 | 0.00012 |
| grok-4-1-fast-reasoning | overall | 60/60 | 0.46 | 0.82 | 0.38 | 0.67 | 32.07 | 0.00012 |
| grok-4-1-fast-reasoning | latex | 30/30 | 0.49 | 0.89 | 0.00 | 0.40 | 30.93 | 0.00012 |
| grok-4-1-fast-reasoning | markdown | 30/30 | 0.44 | 0.78 | 0.67 | 1.00 | 33.21 | 0.00012 |

## C.2 Unstructured documents

**Two-pass LLM extraction with canonical paragraphs.** For unstructured documents we use a two-pass LLM pipeline but keep paragraph boundaries deterministic. Pass 1 extracts paragraph anchors from overlapping line windows, and pass 2 groups those paragraphs into sections. To enforce the "no newlines" requirement, we discard LLM paragraph boundaries and instead canonically segment the source text into blank-line paragraph blocks, then snap section spans to those canonical paragraph boundaries. The resulting hierarchy is thus an overlay on the original text: the LLM provides grouping and level metadata, while paragraph spans remain byte-identical to the source.

---

**Procedure 3** Unstructured Two-Pass Hierarchy Extraction with Canonical Paragraphs

---

**Require:** Unstructured text $X$
**Ensure:** Hierarchy $H$
1: Split $X$ into overlapping line windows
2: **Pass 1:** For each window, prompt the LLM for paragraph start/end anchors
3: Resolve anchors to full-text spans; dedupe and drop overlaps
4: Compute canonical paragraph spans by blank-line segmentation
5: Replace LLM paragraph spans with canonical paragraph spans
6: **Pass 2:** Slide over paragraph windows; prompt LLM to group paragraph IDs into sections
7: Convert each section group to a span covering its paragraph range
8: Snap section spans to cover their child paragraphs; drop overlaps
9: Fill uncovered paragraphs with synthetic sections if needed
10: Build hierarchy nodes from sections and paragraphs; refresh content from spans
11: **return** $H$

---

We evaluate localized paragraph edits on nine research papers (one repetition each, `gpt-4.1`, anchors-two-pass, 150/50 line windows). The target paragraph is selected deterministically from the raw text (hash index over blank-line blocks). We then (i) execute a paragraph-level replace using the hierarchy, (ii) run a full-document baseline with the same replacement, configured to return the full document inside tags with an explicit output-token cap sized to the document length (capped at 50k), and (iii) run a ReAct+`apply_-patch` comparator (Section 5.5) on the same target paragraph and replacement, using the existing 12-step budget and `gpt-4.1`. Success requires an exact match to the expected output with zero outside-target diffs. Table 13 reports per-document results. In this nine-document pilot, the hierarchy-based edit has exact-match success on 9/9 documents. The full-document baseline returns no exact matches (two runs exceed the model's 32k completion limit and are rejected, and the remaining runs either truncate or introduce extra edits) and is included primarily as a feasibility data point under provider completion limits rather than as a quality comparator. ReAct+`apply_patch` reaches 2/9: eight of nine runs terminate at the parse-error budget without applying any patch (outside-target diff = 0 in those rows because no patch is applied), and the one run that did finish (on the ReAct paper itself, Yao et al., 2023) wrote a patch that changed 252 bytes outside the target paragraph—the only observed side-effect case in this pilot. This contrast illustrates two things: the value of deterministic paragraph anchoring for unstructured inputs, and the fragility of patch-based agentic editing on long unstructured prose where the agent has to navigate, anchor, and emit a context-matching patch in a JSON-action loop. The pilot does not establish broad extraction reliability for all unstructured documents.

Table 13: Unstructured localized edit pilot with canonical paragraph boundaries (9 papers, 1 repetition, `gpt-4.1`). The full-document baseline is capped at 50k output tokens, but OpenAI enforces a 32k completion limit, so the longest papers cannot return a full document. The ReAct+`apply_patch` comparator uses a 12-step budget; "Outside diff (react) = 0" rows correspond to runs that terminated at the parse-error budget without applying any patch.

| Document | Target chars (non-ws) | Chunk fails | Hierarchy edit | Outside diff (hier) | Baseline edit | Outside diff (baseline) | ReAct edit | Outside diff (react) |
|---|---|---|---|---|---|---|---|---|
| CodeBERT | 3,393 | 0 | 1 | 0 | 0 | 37,988 | 1 | 0 |
| GraphCodeBERT | 3,182 | 0 | 1 | 0 | 0 | 59,578 | 0 | 0 |
| Hierarchical Transformers | 3,729 | 0 | 1 | 0 | 0 | 20,656 | 0 | 0 |
| Knowledge Editing Survey | 4,363 | 4 | 1 | 0 | 0 | — | 0 | 0 |
| Longformer | 3,949 | 3 | 1 | 0 | 0 | 62,372 | 0 | 0 |
| Position Editing Safety Risks | 4,311 | 1 | 1 | 0 | 0 | 67,687 | 0 | 0 |
| ReAct | 1,559 | 3 | 1 | 0 | 0 | 108,035 | 0 | 252 |
| Safeguarding LLMs Survey | 7,206 | 4 | 1 | 0 | 0 | — | 1 | 0 |
| Toolformer | 3,575 | 2 | 1 | 0 | 0 | 64,101 | 0 | 0 |

# D    Whitespace Handling

## D.1    Whitespace handling and canonicalization

Whitespace is treated as payload in the executor and is never normalized or trimmed during execution. The only optional modification is paragraph-separator insertion when adjacent paragraph nodes would otherwise merge after structural edits. Whitespace is not directly addressable as a standalone target; extra inter-paragraph whitespace is treated as content and can be removed explicitly (e.g., "remove the extra space between P2 and P3"). This section documents the representation and its tradeoffs.

**Canonicalization and payload ownership.**    We use a canonicalization $C(X)$ that preserves document bytes: render the hierarchy deterministically and require the resulting text to match the input (lossless render). For a node $n$ with ordered children $c_1, \ldots, c_k$, the rendered text is

$$\mathrm{render}(n) = \mathrm{prefix}(n) \parallel \mathrm{render}(c_1) \parallel \Delta_1 \parallel \mathrm{render}(c_2) \parallel \cdots \parallel \Delta_{k-1} \parallel \mathrm{render}(c_k) \parallel \Delta_k,$$

where $\mathrm{prefix}(n)$ is the text before the first child, and each $\Delta_i$ is the inter-child or trailing whitespace segment drawn from the original text. We represent each $\Delta_i$ as an explicit whitespace node with label `gap`. Gap nodes are non-movable and non-anchorable, and they are included in the payload footprint so any change is tracked. This representation is lossless under $C(X)$ and makes whitespace ownership explicit.

**Paragraph separators.**    Paragraph boundaries are detected by blank lines (`\n[ \t]*\n`). After sequential plan execution (structural edits), if two paragraph nodes become adjacent and no blank line remains between them, the default `preserve_paragraph_boundaries` policy inserts the minimal separator (typically `\n\n`, or `\n` if already newline-terminated). This insertion is deterministic and expands the payload footprint to include the affected paragraph nodes. A strict-raw mode disables this insertion when byte identity of the original input must be preserved, at the cost of possible paragraph merges after moves.

**Examples and tradeoffs.**

- **Missing trailing separator.** Text: `P1\n\nP2` (no trailing blank line after `P2`). Moving `P2` before `P1` creates an adjacent pair with no separator. With `preserve_paragraph_boundaries`, the executor inserts a minimal blank line and returns `P2\n\nP1\n\n`. In strict-raw mode, the boundary may collapse.

- **Extra blank lines between paragraphs.** Text: `P1\n\n\n\nP2\n\n`. The extra spacing is stored in the gap node between the siblings; moving `P2` elsewhere leaves that extra gap behind, while the paragraph-boundary separator remains enforced at the new location.

- **Leading/trailing whitespace.** Any preamble before the first root and trailing whitespace after the last root are represented as gap nodes so that moving the first/last semantic node does not drag file headers, footers, or trailing padding into a different location.

**Safety interaction.**    Whitespace changes are treated like other payload changes: boundary gap nodes are always included in the payload footprint, and the neighboring boundary leaves are included as a conservative halo because separator placement can touch their local payload. Optional normalization policies (B–D) only affect payload comparison during safety checks; they do not modify stored text. This keeps the executor deterministic and ensures that whitespace-sensitive formats (code, LaTeX) remain byte-identical unless an explicit separator insertion is enabled.

**Whitespace halo examples.**

- **Leading gap + root insert.** Text begins with a blank-line preamble. Inserting a new root at position 0 leaves the leading gap in place, which effectively reattaches that gap to the new root. The gap node must be part of the payload footprint to avoid a false non-interference failure.

- **Nested subsection + trailing insert.** A section contains a nested subsection; inserting a paragraph after the subsection may transfer the boundary newline from the subsection's last paragraph into the new paragraph's boundary. The boundary leaf (last paragraph inside the subsection) must be inside the footprint halo so reconciliation allows the whitespace shift.

# E  Supplementary Diagnostics

This appendix reports selected diagnostics referenced in the main text. Tables 14 and 15 report compound-operation diagnostics; Tables 16 and 17 report policy-brief structural permutation; Tables 18–23 report LaTeX micro-edit details; Tables 24, 25, 26, and 27 report LaTeX structural diagnostics; Tables 28 and 34 report supplementary multi-file refactor variants; Tables 35 and 36 report ReAct per-condition details; and Tables 37 and 38 report worker-output guardrail ablations.

## E.1  Compound Operation Diagnostics

Table 14: Compound operations (10k briefs).

| Method | Success Rate (WS) | Success Rate (Strict) | Side-Effect Rate (WS) | Side-Effect Rate (Strict) | Δ Strict Success | $p$ (Strict Succ) | $p$ (Strict SE) | Time (s) |
|---|---|---|---|---|---|---|---|---|
| gpt-4.1 (Flat) | 0.00 [0.00, 0.20] | 0.00 [0.00, 0.20] | 1.00 [0.80, 1.00] | 1.00 [0.80, 1.00] | – | – | – | 6.00 |
| gpt-4.1 (Functional Safety) | 1.00 [0.80, 1.00] | 1.00 [0.80, 1.00] | 0.00 [0.00, 0.20] | 0.00 [0.00, 0.20] | +1.00 [+0.59, +1.00] | $< 10^{-3}$ | $< 10^{-3}$ | 3.71 |
| grok-4-1-fast-reasoning (Flat) | 0.00 [0.00, 0.20] | 0.00 [0.00, 0.20] | 1.00 [0.80, 1.00] | 1.00 [0.80, 1.00] | – | – | – | 3.37 |
| grok-4-1-fast-reasoning (Functional Safety) | 0.00 [0.00, 0.20] | 0.00 [0.00, 0.20] | 0.00 [0.00, 0.20] | 0.00 [0.00, 0.20] | +0.00 [-0.20, +0.20] | 1.000 | $< 10^{-3}$ | 3.31 |

Table 15: Compound operations (20k briefs).

| Method | Success Rate (WS) | Success Rate (Strict) | Side-Effect Rate (WS) | Side-Effect Rate (Strict) | Δ Strict Success | $p$ (Strict Succ) | $p$ (Strict SE) | Time (s) |
|---|---|---|---|---|---|---|---|---|
| gpt-4.1 (Flat) | 0.00 [0.00, 0.20] | 0.00 [0.00, 0.20] | 1.00 [0.80, 1.00] | 1.00 [0.80, 1.00] | – | – | – | 6.43 |
| gpt-4.1 (Functional Safety) | 1.00 [0.80, 1.00] | 1.00 [0.80, 1.00] | 0.00 [0.00, 0.20] | 0.00 [0.00, 0.20] | +1.00 [+0.59, +1.00] | $< 10^{-3}$ | $< 10^{-3}$ | 3.47 |
| grok-4-1-fast-reasoning (Flat) | 0.00 [0.00, 0.20] | 0.00 [0.00, 0.20] | 1.00 [0.80, 1.00] | 1.00 [0.80, 1.00] | – | – | – | 3.67 |
| grok-4-1-fast-reasoning (Functional Safety) | 0.00 [0.00, 0.20] | 0.00 [0.00, 0.20] | 0.00 [0.00, 0.20] | 0.00 [0.00, 0.20] | +0.00 [-0.20, +0.20] | 1.000 | $< 10^{-3}$ | 3.43 |

## E.2  Policy Brief Structural Permutation

Table 16: Pure section permutation (10k briefs).

| Method | Success Rate (WS) | Success Rate (Strict) | Side-Effect Rate (WS) | Side-Effect Rate (Strict) | Δ Strict Success | $p$ (Strict Succ) | $p$ (Strict SE) | Time (s) |
|---|---|---|---|---|---|---|---|---|
| gpt-4.1 (Flat) | 0.00 [0.00, 0.20] | 0.00 [0.00, 0.20] | 1.00 [0.80, 1.00] | 1.00 [0.80, 1.00] | – | – | – | 7.62 |
| gpt-4.1 (Functional Safety) | 1.00 [0.80, 1.00] | 1.00 [0.80, 1.00] | 0.00 [0.00, 0.20] | 0.00 [0.00, 0.20] | +1.00 [+0.59, +1.00] | $< 10^{-3}$ | $< 10^{-3}$ | 0.96 |
| grok-4-1-fast-reasoning (Flat) | 0.00 [0.00, 0.20] | 0.00 [0.00, 0.20] | 1.00 [0.80, 1.00] | 1.00 [0.80, 1.00] | – | – | – | 3.34 |
| grok-4-1-fast-reasoning (Functional Safety) | 0.00 [0.00, 0.20] | 0.00 [0.00, 0.20] | 0.00 [0.00, 0.20] | 0.00 [0.00, 0.20] | +0.00 [-0.20, +0.20] | 1.000 | $< 10^{-3}$ | 3.33 |

Table 17: Pure section permutation (20k briefs).

| Method | Success Rate (WS) | Success Rate (Strict) | Side-Effect Rate (WS) | Side-Effect Rate (Strict) | Δ Strict Success | $p$ (Strict Succ) | $p$ (Strict SE) | Time (s) |
|---|---|---|---|---|---|---|---|---|
| gpt-4.1 (Flat) | 0.00 [0.00, 0.20] | 0.00 [0.00, 0.20] | 1.00 [0.80, 1.00] | 1.00 [0.80, 1.00] | – | – | – | 5.67 |
| gpt-4.1 (Functional Safety) | 1.00 [0.80, 1.00] | 1.00 [0.80, 1.00] | 0.00 [0.00, 0.20] | 0.00 [0.00, 0.20] | +1.00 [+0.59, +1.00] | $< 10^{-3}$ | $< 10^{-3}$ | 0.94 |
| grok-4-1-fast-reasoning (Flat) | 0.00 [0.00, 0.20] | 0.00 [0.00, 0.20] | 1.00 [0.80, 1.00] | 1.00 [0.80, 1.00] | – | – | – | 3.43 |
| grok-4-1-fast-reasoning (Functional Safety) | 0.00 [0.00, 0.20] | 0.00 [0.00, 0.20] | 0.00 [0.00, 0.20] | 0.00 [0.00, 0.20] | +0.00 [-0.20, +0.20] | 1.000 | $< 10^{-3}$ | 3.32 |

### E.3 LaTeX Micro-Edit Details

Table 18: LaTeX Math Edit (10k). Functional Safety updates the math symbol while preserving surrounding content.

| Method | Success Rate (WS) | Success Rate (Strict) | Side-Effect Rate (WS) | Side-Effect Rate (Strict) | Δ Strict Success | p (Strict Succ) | p (Strict SE) | Time (s) |
|---|---|---|---|---|---|---|---|---|
| gpt-4.1 (Flat) | 0.00 [0.00, 0.07] | 0.00 [0.00, 0.07] | 1.00 [0.93, 1.00] | 1.00 [0.93, 1.00] | – | – | – | 6.28 |
| gpt-4.1 (Functional Safety) | 1.00 [0.93, 1.00] | 1.00 [0.93, 1.00] | 0.00 [0.00, 0.07] | 0.00 [0.00, 0.07] | +1.00 [+0.86, +1.00] | $< 10^{-3}$ | $< 10^{-3}$ | 3.41 |
| grok-4-1-fast-reasoning (Flat) | 0.00 [0.00, 0.07] | 0.00 [0.00, 0.07] | 1.00 [0.93, 1.00] | 1.00 [0.93, 1.00] | – | – | – | 3.43 |
| grok-4-1-fast-reasoning (Functional Safety) | 0.70 [0.56, 0.81] | 0.70 [0.56, 0.81] | 0.00 [0.00, 0.07] | 0.00 [0.00, 0.07] | +0.70 [+0.49, +0.81] | $< 10^{-3}$ | $< 10^{-3}$ | 5.44 |

Table 19: LaTeX Math Edit (20k). Functional Safety updates the math symbol while preserving surrounding content.

| Method | Success Rate (WS) | Success Rate (Strict) | Side-Effect Rate (WS) | Side-Effect Rate (Strict) | Δ Strict Success | p (Strict Succ) | p (Strict SE) | Time (s) |
|---|---|---|---|---|---|---|---|---|
| gpt-4.1 (Flat) | 0.00 [0.00, 0.07] | 0.00 [0.00, 0.07] | 1.00 [0.93, 1.00] | 1.00 [0.93, 1.00] | – | – | – | 6.15 |
| gpt-4.1 (Functional Safety) | 1.00 [0.93, 1.00] | 1.00 [0.93, 1.00] | 0.00 [0.00, 0.07] | 0.00 [0.00, 0.07] | +1.00 [+0.86, +1.00] | $< 10^{-3}$ | $< 10^{-3}$ | 5.03 |
| grok-4-1-fast-reasoning (Flat) | 0.00 [0.00, 0.07] | 0.00 [0.00, 0.07] | 1.00 [0.93, 1.00] | 1.00 [0.93, 1.00] | – | – | – | 2.66 |
| grok-4-1-fast-reasoning (Functional Safety) | 0.70 [0.56, 0.81] | 0.70 [0.56, 0.81] | 0.00 [0.00, 0.07] | 0.00 [0.00, 0.07] | +0.70 [+0.49, +0.81] | $< 10^{-3}$ | $< 10^{-3}$ | 6.82 |

Table 20: LaTeX Cross-Reference Update (10k). Functional Safety updates the label and matching \ref while preserving surrounding content.

| Method | Success Rate (WS) | Success Rate (Strict) | Side-Effect Rate (WS) | Side-Effect Rate (Strict) | Δ Strict Success | p (Strict Succ) | p (Strict SE) | Time (s) |
|---|---|---|---|---|---|---|---|---|
| gpt-4.1 (Flat) | 0.00 [0.00, 0.07] | 0.00 [0.00, 0.07] | 1.00 [0.93, 1.00] | 1.00 [0.93, 1.00] | – | – | – | 6.25 |
| gpt-4.1 (Functional Safety) | 1.00 [0.93, 1.00] | 1.00 [0.93, 1.00] | 0.00 [0.00, 0.07] | 0.00 [0.00, 0.07] | +1.00 [+0.86, +1.00] | $< 10^{-3}$ | $< 10^{-3}$ | 2.18 |
| grok-4-1-fast-reasoning (Flat) | 0.00 [0.00, 0.07] | 0.00 [0.00, 0.07] | 1.00 [0.93, 1.00] | 1.00 [0.93, 1.00] | – | – | – | 5.28 |
| grok-4-1-fast-reasoning (Functional Safety) | 0.70 [0.56, 0.81] | 0.70 [0.56, 0.81] | 0.00 [0.00, 0.07] | 0.00 [0.00, 0.07] | +0.70 [+0.49, +0.81] | $< 10^{-3}$ | $< 10^{-3}$ | 6.71 |

Table 21: LaTeX Cross-Reference Update (20k). Functional Safety updates the label and matching \ref while preserving surrounding content.

| Method | Success Rate (WS) | Success Rate (Strict) | Side-Effect Rate (WS) | Side-Effect Rate (Strict) | Δ Strict Success | p (Strict Succ) | p (Strict SE) | Time (s) |
|---|---|---|---|---|---|---|---|---|
| gpt-4.1 (Flat) | 0.00 [0.00, 0.07] | 0.00 [0.00, 0.07] | 1.00 [0.93, 1.00] | 1.00 [0.93, 1.00] | – | – | – | 6.71 |
| gpt-4.1 (Functional Safety) | 1.00 [0.93, 1.00] | 1.00 [0.93, 1.00] | 0.00 [0.00, 0.07] | 0.00 [0.00, 0.07] | +1.00 [+0.86, +1.00] | $< 10^{-3}$ | $< 10^{-3}$ | 2.52 |
| grok-4-1-fast-reasoning (Flat) | 0.00 [0.00, 0.07] | 0.00 [0.00, 0.07] | 1.00 [0.93, 1.00] | 1.00 [0.93, 1.00] | – | – | – | 3.88 |
| grok-4-1-fast-reasoning (Functional Safety) | 0.70 [0.56, 0.81] | 0.70 [0.56, 0.81] | 0.00 [0.00, 0.07] | 0.00 [0.00, 0.07] | +0.70 [+0.49, +0.81] | $< 10^{-3}$ | $< 10^{-3}$ | 6.21 |

Table 22: LaTeX Citation Update (10k). Functional Safety updates the citation key while preserving surrounding content.

| Method | Success Rate (WS) | Success Rate (Strict) | Side-Effect Rate (WS) | Side-Effect Rate (Strict) | Δ Strict Success | p (Strict Succ) | p (Strict SE) | Time (s) |
|---|---|---|---|---|---|---|---|---|
| gpt-4.1 (Flat) | 0.00 [0.00, 0.07] | 0.00 [0.00, 0.07] | 1.00 [0.93, 1.00] | 1.00 [0.93, 1.00] | – | – | – | 5.70 |
| gpt-4.1 (Functional Safety) | 1.00 [0.93, 1.00] | 1.00 [0.93, 1.00] | 0.00 [0.00, 0.07] | 0.00 [0.00, 0.07] | +1.00 [+0.86, +1.00] | $< 10^{-3}$ | $< 10^{-3}$ | 2.56 |
| grok-4-1-fast-reasoning (Flat) | 0.00 [0.00, 0.07] | 0.00 [0.00, 0.07] | 1.00 [0.93, 1.00] | 1.00 [0.93, 1.00] | – | – | – | 3.66 |
| grok-4-1-fast-reasoning (Functional Safety) | 0.70 [0.56, 0.81] | 0.70 [0.56, 0.81] | 0.00 [0.00, 0.07] | 0.00 [0.00, 0.07] | +0.70 [+0.49, +0.81] | $< 10^{-3}$ | $< 10^{-3}$ | 9.98 |

Table 23: LaTeX Citation Update (20k). Functional Safety updates the citation key while preserving surrounding content.

| Method | Success Rate (WS) | Success Rate (Strict) | Side-Effect Rate (WS) | Side-Effect Rate (Strict) | Δ Strict Success | p (Strict Succ) | p (Strict SE) | Time (s) |
|---|---|---|---|---|---|---|---|---|
| gpt-4.1 (Flat) | 0.00 [0.00, 0.07] | 0.00 [0.00, 0.07] | 1.00 [0.93, 1.00] | 1.00 [0.93, 1.00] | – | – | – | 7.63 |
| gpt-4.1 (Functional Safety) | 1.00 [0.93, 1.00] | 1.00 [0.93, 1.00] | 0.00 [0.00, 0.07] | 0.00 [0.00, 0.07] | +1.00 [+0.86, +1.00] | $< 10^{-3}$ | $< 10^{-3}$ | 2.94 |
| grok-4-1-fast-reasoning (Flat) | 0.00 [0.00, 0.07] | 0.00 [0.00, 0.07] | 1.00 [0.93, 1.00] | 1.00 [0.93, 1.00] | – | – | – | 2.99 |
| grok-4-1-fast-reasoning (Functional Safety) | 0.70 [0.56, 0.81] | 0.70 [0.56, 0.81] | 0.00 [0.00, 0.07] | 0.00 [0.00, 0.07] | +0.70 [+0.49, +0.81] | $< 10^{-3}$ | $< 10^{-3}$ | 9.94 |

## E.4 LaTeX Structural Diagnostics

Table 24: LaTeX section permutation (10k).

| Method | Success Rate (WS) | Success Rate (Strict) | Side-Effect Rate (WS) | Side-Effect Rate (Strict) | Time (s) |
|---|---|---|---|---|---|
| gpt-4.1 (Flat) | 0.00 [0.00, 0.20] | 0.00 [0.00, 0.20] | 1.00 [0.80, 1.00] | 1.00 [0.80, 1.00] | 6.87 |
| gpt-4.1 (ReAct+patch) | 0.00 [0.00, 0.20] | 0.00 [0.00, 0.20] | 0.00 [0.00, 0.20] | 0.00 [0.00, 0.20] | 12.54 |
| gpt-4.1 (Functional Safety) | 1.00 [0.80, 1.00] | 1.00 [0.80, 1.00] | 0.00 [0.00, 0.20] | 0.00 [0.00, 0.20] | 1.10 |
| grok-4-1-fast-reasoning (Flat) | 0.00 [0.00, 0.20] | 0.00 [0.00, 0.20] | 1.00 [0.80, 1.00] | 1.00 [0.80, 1.00] | 7.45 |
| grok-4-1-fast-reasoning (ReAct+patch) | 0.00 [0.00, 0.20] | 0.00 [0.00, 0.20] | 0.00 [0.00, 0.20] | 0.00 [0.00, 0.20] | 383.21 |
| grok-4-1-fast-reasoning (Functional Safety) | 1.00 [0.80, 1.00] | 1.00 [0.80, 1.00] | 0.00 [0.00, 0.20] | 0.00 [0.00, 0.20] | 4.21 |

Table 25: LaTeX section permutation (20k).

| Method | Success Rate (WS) | Success Rate (Strict) | Side-Effect Rate (WS) | Side-Effect Rate (Strict) | Time (s) |
|---|---|---|---|---|---|
| gpt-4.1 (Flat) | 0.00 [0.00, 0.20] | 0.00 [0.00, 0.20] | 1.00 [0.80, 1.00] | 1.00 [0.80, 1.00] | 11.52 |
| gpt-4.1 (ReAct+patch) | 0.00 [0.00, 0.20] | 0.00 [0.00, 0.20] | 0.00 [0.00, 0.20] | 0.00 [0.00, 0.20] | 10.92 |
| gpt-4.1 (Functional Safety) | 1.00 [0.80, 1.00] | 1.00 [0.80, 1.00] | 0.00 [0.00, 0.20] | 0.00 [0.00, 0.20] | 1.49 |
| grok-4-1-fast-reasoning (Flat) | 0.00 [0.00, 0.20] | 0.00 [0.00, 0.20] | 1.00 [0.80, 1.00] | 1.00 [0.80, 1.00] | 5.76 |
| grok-4-1-fast-reasoning (ReAct+patch) | 0.00 [0.00, 0.20] | 0.00 [0.00, 0.20] | 0.00 [0.00, 0.20] | 0.00 [0.00, 0.20] | 337.69 |
| grok-4-1-fast-reasoning (Functional Safety) | 1.00 [0.80, 1.00] | 1.00 [0.80, 1.00] | 0.00 [0.00, 0.20] | 0.00 [0.00, 0.20] | 3.66 |

Table 26: LaTeX section relocation (10k).

| Method | Success Rate (WS) | Success Rate (Strict) | Side-Effect Rate (WS) | Side-Effect Rate (Strict) | Time (s) |
|---|---|---|---|---|---|
| gpt-4.1 (Flat) | 0.00 [0.00, 0.20] | 0.00 [0.00, 0.20] | 1.00 [0.80, 1.00] | 1.00 [0.80, 1.00] | 10.71 |
| gpt-4.1 (ReAct+patch) | 0.00 [0.00, 0.20] | 0.00 [0.00, 0.20] | 1.00 [0.80, 1.00] | 1.00 [0.80, 1.00] | 22.43 |
| gpt-4.1 (Functional Safety) | 1.00 [0.80, 1.00] | 1.00 [0.80, 1.00] | 0.00 [0.00, 0.20] | 0.00 [0.00, 0.20] | 4.21 |
| grok-4-1-fast-reasoning (Flat) | 0.00 [0.00, 0.20] | 0.00 [0.00, 0.20] | 1.00 [0.80, 1.00] | 1.00 [0.80, 1.00] | 7.36 |
| grok-4-1-fast-reasoning (ReAct+patch) | 0.07 [0.01, 0.30] | 0.07 [0.01, 0.30] | 0.93 [0.70, 0.99] | 0.93 [0.70, 0.99] | 267.70 |
| grok-4-1-fast-reasoning (Functional Safety) | 1.00 [0.80, 1.00] | 1.00 [0.80, 1.00] | 0.00 [0.00, 0.20] | 0.00 [0.00, 0.20] | 12.47 |

Table 27: LaTeX section relocation (20k).

| Method | Success Rate (WS) | Success Rate (Strict) | Side-Effect Rate (WS) | Side-Effect Rate (Strict) | Time (s) |
|---|---|---|---|---|---|
| gpt-4.1 (Flat) | 0.00 [0.00, 0.20] | 0.00 [0.00, 0.20] | 1.00 [0.80, 1.00] | 1.00 [0.80, 1.00] | 8.82 |
| gpt-4.1 (ReAct+patch) | 0.00 [0.00, 0.20] | 0.00 [0.00, 0.20] | 1.00 [0.80, 1.00] | 1.00 [0.80, 1.00] | 24.65 |
| gpt-4.1 (Functional Safety) | 1.00 [0.80, 1.00] | 1.00 [0.80, 1.00] | 0.00 [0.00, 0.20] | 0.00 [0.00, 0.20] | 8.64 |
| grok-4-1-fast-reasoning (Flat) | 0.00 [0.00, 0.20] | 0.00 [0.00, 0.20] | 1.00 [0.80, 1.00] | 1.00 [0.80, 1.00] | 6.62 |
| grok-4-1-fast-reasoning (ReAct+patch) | 0.00 [0.00, 0.20] | 0.00 [0.00, 0.20] | 0.93 [0.70, 0.99] | 0.93 [0.70, 0.99] | 244.19 |
| grok-4-1-fast-reasoning (Functional Safety) | 1.00 [0.80, 1.00] | 1.00 [0.80, 1.00] | 0.00 [0.00, 0.20] | 0.00 [0.00, 0.20] | 12.70 |

## E.5 Supplementary Multi-File Refactor Results

Table 28: Multi-file refactor setup variants. "Exact" uses the original package layout; "open" asks the planner to propose a new layout.

| Case | Monolith | Layout target | Prompt constraint | Tests |
|---|---|---|---|---|
| compute_refactor | concat | exact | reconstruct original tree | compute suite (149 tests) |
| vector_refactor | concat | exact | reconstruct original tree | vector suite (45 tests, 1 skipped) |
| compute_refactor_imports_top | imports_top | exact | reconstruct original tree | compute suite (149 tests) |
| vector_refactor_imports_top | imports_top | exact | reconstruct original tree | vector suite (45 tests, 1 skipped) |
| compute_refactor_open_relaxed | concat | open | behavior-only layout suggestion | compute suite (149 tests) |
| compute_refactor_open_compat | concat | open | add import-path compatibility | compute suite (149 tests) |
| vector_refactor_open_relaxed | concat | open | behavior-only layout suggestion | vector suite (45 tests, 1 skipped) |
| vector_refactor_open_compat | concat | open | add import-path compatibility | vector suite (45 tests, 1 skipped) |

Table 29: Multi-file Python compute refactor with consolidated imports at the top of the monolith.

| Approach | Runs | Failed | Layout Match | Byte Id | WS Id | TeX Id | AST Id | Sig Δ | Import Δ | Repair Rounds | Time (s) | Test Pass |
|---|---|---|---|---|---|---|---|---|---|---|---|---|
| claude_cli_reference | 5 | 0 | 5 | 0.00+/-0.00 | 0.00+/-0.00 | – | 0.12+/-0.18 | 3.2+/-3.3 | 15.4+/-21.3/11.8+/-6.2 | – | 1188.39+/-115.97 | 1.00+/-0.00 |
| codex_cli_default_medium | 5 | 0 | 5 | 0.00+/-0.00 | 0.00+/-0.00 | – | 0.00+/-0.00 | 0.0+/-0.0 | 2.8+/-0.4/3.0+/-0.0 | – | 538.44+/-156.49 | 1.00+/-0.00 |
| hier_gpt-4o | 5 | 0 | 5 | 0.00+/-0.00 | 0.00+/-0.00 | – | 0.00+/-0.00 | 0.0+/-0.0 | 146.8+/-23.4/57.8+/-23.4 | 1.6+/-0.5 | 4.75+/-1.18 | 1.00+/-0.00 |
| hier_gpt-5 | 5 | 0 | 5 | 0.00+/-0.00 | 0.00+/-0.00 | – | 0.00+/-0.00 | 0.0+/-0.0 | 178.0+/-0.0/89.0+/-0.0 | 1.0+/-0.0 | 54.97+/-2.82 | 1.00+/-0.00 |
| hier_grok-4-1-fast-reasoning | 5 | 0 | 5 | 0.00+/-0.00 | 0.00+/-0.00 | – | 0.00+/-0.00 | 0.0+/-0.0 | 149.2+/-26.3/60.2+/-26.3 | 1.0+/-0.0 | 60.86+/-7.66 | 1.00+/-0.00 |

Table 30: Multi-file Python vector refactor with consolidated imports at the top of the monolith.

| Approach | Runs | Failed | Layout Match | Byte Id | WS Id | TeX Id | AST Id | Sig Δ | Import Δ | Repair Rounds | Time (s) | Test Pass |
|---|---|---|---|---|---|---|---|---|---|---|---|---|
| claude_cli_reference | 5 | 0 | 4 | 0.03+/-0.05 | 0.03+/-0.05 | – | 0.07+/-0.05 | 16.2+/-24.6 | 42.6+/-42.6/55.0+/-36.0 | – | 1861.38+/-1559.05 | 0.97+/-0.01 |
| codex_cli_default_medium | 5 | 0 | 5 | 0.00+/-0.00 | 0.00+/-0.00 | – | 0.00+/-0.00 | 0.2+/-0.4 | 109.8+/-56.7/115.8+/-91.8 | – | 1888.04+/-349.87 | 0.99+/-0.01 |
| hier_gpt-4o | 5 | 0 | 5 | 0.00+/-0.00 | 0.00+/-0.00 | – | 0.00+/-0.00 | 0.0+/-0.0 | 526.8+/-25.8/187.8+/-25.8 | 11.0+/-1.9 | 30.06+/-15.38 | 0.98+/-0.00 |
| hier_gpt-5 | 4 | 0 | 4 | 0.00+/-0.00 | 0.00+/-0.00 | – | 0.00+/-0.00 | 0.0+/-0.0 | 554.0+/-0.0/215.0+/-0.0 | 10.8+/-2.9 | 660.84+/-82.91 | 0.98+/-0.01 |
| hier_grok-4-1-fast-reasoning | 5 | 0 | 5 | 0.00+/-0.00 | 0.00+/-0.00 | – | 0.00+/-0.00 | 0.0+/-0.0 | 517.0+/-20.1/178.0+/-20.1 | 9.0+/-5.6 | 311.74+/-167.36 | 0.98+/-0.00 |

Table 31: Open-layout compute refactor (relaxed). The planner proposes a layout; identity metrics are reported against the original tree for reference.

| Approach | Runs | Failed | Layout Match | Byte Id | WS Id | TeX Id | AST Id | Sig Δ | Import Δ | Repair Rounds | Time (s) | Test Pass |
|---|---|---|---|---|---|---|---|---|---|---|---|---|
| claude_cli_reference | 5 | 0 | 0 | 0.85+/-0.05 | 0.85+/-0.05 | – | 0.92+/-0.06 | 1.0+/-2.2 | 3.0+/-5.6/5.8+/-11.9 | – | 1296.86+/-57.99 | 0.00+/-0.00 |
| codex_cli_default_medium | 5 | 0 | 0 | 0.00+/-0.00 | 0.00+/-0.00 | – | 0.00+/-0.00 | 0.0+/-0.0 | 0.0+/-0.0/0.0+/-0.0 | – | 271.57+/-45.50 | 0.20+/-0.45 |
| hier_gpt-4o | 5 | 0 | 0 | 0.15+/-0.09 | 0.15+/-0.09 | – | 0.62+/-0.37 | 0.0+/-0.0 | 2.4+/-3.3/6.6+/-12.6 | 0.6+/-0.5 | 3.42+/-0.96 | 0.00+/-0.00 |
| hier_gpt-5 | 5 | 0 | 0 | 0.14+/-0.19 | 0.14+/-0.19 | – | 0.33+/-0.45 | 0.0+/-0.0 | 0.0+/-0.0/22.8+/-31.2 | 0.2+/-0.4 | 34.51+/-13.77 | 0.20+/-0.45 |
| hier_grok-4-1-fast-reasoning | 5 | 0 | 0 | 0.05+/-0.10 | 0.05+/-0.10 | – | 0.20+/-0.45 | 0.0+/-0.0 | 0.0+/-0.0/0.0+/-0.0 | 0.0+/-0.0 | 22.34+/-7.07 | 0.00+/-0.00 |

Table 32: Open-layout compute refactor (compat). The prompt requests compatibility with prior import paths and public exports.

| Approach | Runs | Failed | Layout Match | Byte Id | WS Id | TeX Id | AST Id | Sig Δ | Import Δ | Repair Rounds | Time (s) | Test Pass |
|---|---|---|---|---|---|---|---|---|---|---|---|---|
| claude_cli_reference | 5 | 0 | 0 | 0.83+/-0.13 | 0.83+/-0.13 | – | 0.90+/-0.14 | 5.2+/-11.6 | 5.6+/-8.0/6.2+/-11.1 | – | 1320.36+/-212.39 | 0.00+/-0.00 |
| codex_cli_default_medium | 5 | 0 | 0 | 0.00+/-0.00 | 0.00+/-0.00 | – | 0.00+/-0.00 | 0.0+/-0.0 | 0.0+/-0.0/0.0+/-0.0 | – | 412.92+/-142.06 | 0.00+/-0.00 |
| hier_gpt-4o | 5 | 0 | 0 | 0.08+/-0.05 | 0.08+/-0.05 | – | 0.33+/-0.20 | 0.0+/-0.0 | 3.6+/-3.3/37.0+/-33.7 | 0.2+/-0.4 | 3.05+/-0.92 | 0.00+/-0.00 |
| hier_grok-4-1-fast-reasoning | 4 | 0 | 0 | 0.27+/-0.18 | 0.27+/-0.18 | – | 0.75+/-0.50 | 0.0+/-0.0 | 0.0+/-0.0/0.0+/-0.0 | 0.0+/-0.0 | 30.45+/-12.62 | 0.00+/-0.00 |

Table 33: Open-layout vector refactor (relaxed). The planner proposes a layout; identity metrics are reported against the original tree for reference.

| Approach | Runs | Failed | Layout Match | Byte Id | WS Id | TeX Id | AST Id | Sig Δ | Import Δ | Repair Rounds | Time (s) | Test Pass |
|---|---|---|---|---|---|---|---|---|---|---|---|---|
| claude_cli_reference | 5 | 0 | 0 | 0.04+/-0.04 | 0.04+/-0.04 | – | 0.64+/-0.22 | 0.0+/-0.0 | 30.8+/-29.2/66.6+/-68.9 | – | 1568.02+/-1259.75 | 0.96+/-0.00 |
| codex_cli_default_medium | 5 | 0 | 0 | 0.00+/-0.00 | 0.00+/-0.00 | – | 0.00+/-0.00 | 0.0+/-0.0 | 0.0+/-0.0/0.0+/-0.0 | – | 733.87+/-56.03 | 0.96+/-0.00 |
| hier_gpt-4o | 5 | 0 | 0 | 0.00+/-0.00 | 0.00+/-0.00 | – | 0.00+/-0.00 | 0.0+/-0.0 | 0.0+/-0.0/0.0+/-0.0 | 2.6+/-1.5 | 18.68+/-6.28 | 0.96+/-0.00 |
| hier_gpt-5 | 2 | 0 | 0 | 0.33+/-0.14 | 0.33+/-0.14 | – | 0.96+/-0.06 | 0.0+/-0.0 | 1.0+/-1.4/0.5+/-0.7 | 1.5+/-2.1 | 254.63+/-255.50 | 0.96+/-0.00 |
| hier_grok-4-1-fast-reasoning | 5 | 0 | 0 | 0.05+/-0.04 | 0.05+/-0.04 | – | 0.70+/-0.41 | 0.0+/-0.0 | 0.0+/-0.0/3.8+/-6.1 | 0.2+/-0.4 | 67.79+/-9.19 | 0.96+/-0.00 |

Table 34: Open-layout vector refactor (compat). The prompt requests compatibility with prior import paths and public exports.

| Approach | Runs | Failed | Layout Match | Byte Id | WS Id | TeX Id | AST Id | Sig Δ | Import Δ | Repair Rounds | Time (s) | Test Pass |
|---|---|---|---|---|---|---|---|---|---|---|---|---|
| claude_cli_reference | 5 | 0 | 0 | 0.05+/-0.04 | 0.05+/-0.04 | – | 0.54+/-0.20 | 0.0+/-0.0 | 66.6+/-32.5/56.4+/-30.7 | – | 1303.92+/-1378.88 | 0.96+/-0.00 |
| codex_cli_default_medium | 5 | 0 | 0 | 0.00+/-0.00 | 0.00+/-0.00 | – | 0.00+/-0.00 | 0.0+/-0.0 | 0.0+/-0.0/0.0+/-0.0 | – | 883.09+/-175.52 | 0.96+/-0.00 |
| hier_gpt-4o | 5 | 0 | 0 | 0.00+/-0.00 | 0.00+/-0.00 | – | 0.00+/-0.00 | 0.0+/-0.0 | 0.0+/-0.0/0.0+/-0.0 | 1.0+/-0.7 | 14.25+/-2.48 | 0.96+/-0.00 |
| hier_grok-4-1-fast-reasoning | 5 | 0 | 0 | 0.04+/-0.03 | 0.04+/-0.03 | – | 0.61+/-0.46 | 0.0+/-0.0 | 0.0+/-0.0/118.4+/-184.6 | 0.0+/-0.0 | 77.29+/-14.26 | 0.96+/-0.00 |

### E.6 ReAct Baseline Details

**ReAct + Apply-Patch.**

Table 35: ReAct+`apply_patch` baseline. Rows used in the main summary table use 50 runs per model/condition over 10 document IDs; retained diagnostic rows outside that summary remain at their original coverage and can be read from the finish-denominator column. The patcher follows a `git apply`-style strategy: line numbers are hints and hunks are placed by matching context. `parse_error_limit` reflects repeated JSON violations, and `tool_error_limit` captures malformed patches.

| Task | Size | Model | Success (WS) | Side-Effect (WS) | Steps | Time (s) | Finish |
|---|---|---|---|---|---|---|---|
| code_visibility | small | gpt-4.1 | 0.02 [0.00, 0.10] | 0.52 [0.39, 0.65] | 7.1 | 12.2 | finished (26/50) |
| code_visibility | small | grok-4-1-fast-reasoning | 0.02 [0.00, 0.10] | 0.20 [0.11, 0.33] | 4.9 | 155.9 | tool_error_limit (40/50) |
| code_visibility | medium | gpt-4.1 | 0.00 [0.00, 0.07] | 0.30 [0.19, 0.44] | 7.4 | 12.4 | tool_error_limit (22/50) |
| code_visibility | medium | grok-4-1-fast-reasoning | 0.02 [0.00, 0.10] | 0.14 [0.07, 0.26] | 5.8 | 200.1 | tool_error_limit (42/50) |
| contextual_insert | 2k | gpt-4.1 | 0.96 [0.87, 0.99] | 0.04 [0.01, 0.13] | 5.1 | 6.3 | finished (50/50) |
| contextual_insert | 2k | grok-4-1-fast-reasoning | 0.60 [0.46, 0.72] | 0.38 [0.26, 0.52] | 4.8 | 70.1 | finished (49/50) |
| doc_relocate | 10k | gpt-4.1 | 0.00 [0.00, 0.07] | 0.88 [0.76, 0.94] | 9.6 | 18.0 | parse_error_limit (27/50) |
| doc_relocate | 10k | grok-4-1-fast-reasoning | 0.46 [0.33, 0.60] | 0.52 [0.39, 0.65] | 7.5 | 272.4 | finished (23/50) |
| doc_relocate | 20k | gpt-4.1 | 0.00 [0.00, 0.07] | 0.98 [0.90, 1.00] | 7.9 | 13.0 | parse_error_limit (37/50) |
| doc_relocate | 20k | grok-4-1-fast-reasoning | 0.00 [0.00, 0.07] | 0.42 [0.29, 0.56] | 10.2 | 415.4 | parse_error_limit (34/50) |
| doc_reorder | 10k | gpt-4.1 | 0.00 [0.00, 0.20] | 0.00 [0.00, 0.20] | 9.3 | 23.7 | parse_error_limit (15/15) |
| doc_reorder | 10k | grok-4-1-fast-reasoning | 0.20 [0.07, 0.45] | 0.07 [0.01, 0.30] | 7.9 | 378.0 | parse_error_limit (10/15) |
| doc_reorder | 20k | gpt-4.1 | 0.00 [0.00, 0.20] | 0.07 [0.01, 0.30] | 9.1 | 14.7 | parse_error_limit (14/15) |
| doc_reorder | 20k | grok-4-1-fast-reasoning | 0.00 [0.00, 0.20] | 0.00 [0.00, 0.20] | 10.8 | 614.4 | parse_error_limit (13/15) |
| latex_cite_update | 10k | gpt-4.1 | 0.38 [0.26, 0.52] | 0.62 [0.48, 0.74] | 7.6 | 10.1 | finished (36/50) |
| latex_cite_update | 10k | grok-4-1-fast-reasoning | 0.90 [0.79, 0.96] | 0.10 [0.04, 0.21] | 5.8 | 94.3 | finished (48/50) |
| latex_cite_update | 20k | gpt-4.1 | 0.24 [0.14, 0.37] | 0.76 [0.63, 0.86] | 7.1 | 8.5 | finished (33/50) |
| latex_cite_update | 20k | grok-4-1-fast-reasoning | 0.88 [0.76, 0.94] | 0.12 [0.06, 0.24] | 5.9 | 97.6 | finished (49/50) |
| latex_math_edit | 10k | gpt-4.1 | 1.00 [0.93, 1.00] | 0.00 [0.00, 0.07] | 5.8 | 5.8 | finished (50/50) |
| latex_math_edit | 10k | grok-4-1-fast-reasoning | 0.98 [0.90, 1.00] | 0.02 [0.00, 0.10] | 5.7 | 33.7 | finished (50/50) |
| latex_math_edit | 20k | gpt-4.1 | 0.98 [0.90, 1.00] | 0.02 [0.00, 0.10] | 5.6 | 4.9 | finished (50/50) |
| latex_math_edit | 20k | grok-4-1-fast-reasoning | 1.00 [0.93, 1.00] | 0.00 [0.00, 0.07] | 5.5 | 31.3 | finished (50/50) |
| latex_xref_update | 10k | gpt-4.1 | 0.24 [0.14, 0.37] | 0.76 [0.63, 0.86] | 9.9 | 12.4 | finished (26/50) |
| latex_xref_update | 10k | grok-4-1-fast-reasoning | 0.98 [0.90, 1.00] | 0.02 [0.00, 0.10] | 6.9 | 121.3 | finished (49/50) |
| latex_xref_update | 20k | gpt-4.1 | 0.26 [0.16, 0.40] | 0.74 [0.60, 0.84] | 10.0 | 12.8 | finished (37/50) |
| latex_xref_update | 20k | grok-4-1-fast-reasoning | 0.88 [0.76, 0.94] | 0.12 [0.06, 0.24] | 7.5 | 149.1 | finished (45/50) |
| para_consolidate | 10k | gpt-4.1 | 0.66 [0.52, 0.78] | 0.06 [0.02, 0.16] | 5.0 | 5.4 | finished (49/50) |
| para_consolidate | 10k | grok-4-1-fast-reasoning | 0.80 [0.67, 0.89] | 0.00 [0.00, 0.07] | 5.3 | 152.7 | finished (40/50) |
| para_consolidate | 20k | gpt-4.1 | 1.00 [0.93, 1.00] | 0.00 [0.00, 0.07] | 5.0 | 5.7 | finished (50/50) |
| para_consolidate | 20k | grok-4-1-fast-reasoning | 0.92 [0.81, 0.97] | 0.00 [0.00, 0.07] | 5.8 | 164.8 | finished (47/50) |
| para_edit | 2k | gpt-4.1 | 1.00 [0.93, 1.00] | 0.00 [0.00, 0.07] | 4.0 | 3.9 | finished (50/50) |
| para_edit | 2k | grok-4-1-fast-reasoning | 0.96 [0.87, 0.99] | 0.06 [0.02, 0.16] | 5.1 | 59.8 | finished (50/50) |

**ReAct Basic.**

Table 36: ReAct baseline with raw span-editing tools. Common failure modes include malformed span arguments, raw-index edits that corrupt headers, and repeated invalid tool calls. `tool_error_limit` reflects repeated invalid tool calls; `parse_error_limit` indicates repeated JSON violations.

| Task | Size | Model | Success (WS) | Side-Effect (WS) | Steps | Time (s) | Finish |
|---|---|---|---|---|---|---|---|
| code_visibility | small | gpt-4.1 | 0.00 [0.00, 0.20] | 0.47 [0.25, 0.70] | 7.1 | 7.4 | max_steps (8/15) |
| code_visibility | small | grok-4-1-fast-reasoning | 0.00 [0.00, 0.20] | 0.60 [0.36, 0.80] | 8.0 | 211.1 | max_steps (15/15) |
| code_visibility | medium | gpt-4.1 | 0.00 [0.00, 0.20] | 0.60 [0.36, 0.80] | 7.9 | 8.6 | max_steps (12/15) |
| code_visibility | medium | grok-4-1-fast-reasoning | 0.00 [0.00, 0.20] | 0.53 [0.30, 0.75] | 8.0 | 192.9 | max_steps (15/15) |
| contextual_insert | 2k | gpt-4.1 | 0.20 [0.07, 0.45] | 0.80 [0.55, 0.93] | 5.0 | 5.0 | finished (15/15) |
| contextual_insert | 2k | grok-4-1-fast-reasoning | 1.00 [0.80, 1.00] | 0.00 [0.00, 0.20] | 5.9 | 46.6 | finished (15/15) |
| doc_relocate | 10k | gpt-4.1 | 0.00 [0.00, 0.20] | 1.00 [0.80, 1.00] | 7.6 | 6.3 | max_steps (11/15) |
| doc_relocate | 10k | grok-4-1-fast-reasoning | 0.93 [0.70, 0.99] | 0.07 [0.01, 0.30] | 7.5 | 79.6 | finished (13/15) |
| doc_relocate | 20k | gpt-4.1 | 0.00 [0.00, 0.20] | 1.00 [0.80, 1.00] | 7.5 | 6.1 | finished (9/15) |
| doc_relocate | 20k | grok-4-1-fast-reasoning | 1.00 [0.80, 1.00] | 0.00 [0.00, 0.20] | 7.3 | 81.0 | finished (14/15) |
| doc_reorder | 10k | gpt-4.1 | 0.00 [0.00, 0.20] | 1.00 [0.80, 1.00] | 5.1 | 3.8 | finished (15/15) |
| doc_reorder | 10k | grok-4-1-fast-reasoning | 0.93 [0.70, 0.99] | 0.00 [0.00, 0.20] | 5.2 | 88.0 | finished (14/15) |
| doc_reorder | 20k | gpt-4.1 | 0.00 [0.00, 0.20] | 0.73 [0.48, 0.89] | 4.5 | 3.3 | finished (15/15) |
| doc_reorder | 20k | grok-4-1-fast-reasoning | 1.00 [0.80, 1.00] | 0.00 [0.00, 0.20] | 5.0 | 84.9 | finished (15/15) |
| latex_cite_update | 10k | gpt-4.1 | 0.00 [0.00, 0.20] | 1.00 [0.80, 1.00] | 4.8 | 3.4 | finished (15/15) |
| latex_cite_update | 10k | grok-4-1-fast-reasoning | 1.00 [0.80, 1.00] | 0.00 [0.00, 0.20] | 4.5 | 23.2 | finished (15/15) |
| latex_cite_update | 20k | gpt-4.1 | 0.00 [0.00, 0.20] | 1.00 [0.80, 1.00] | 5.1 | 4.0 | finished (15/15) |
| latex_cite_update | 20k | grok-4-1-fast-reasoning | 0.93 [0.70, 0.99] | 0.07 [0.01, 0.30] | 4.6 | 26.0 | finished (15/15) |
| latex_math_edit | 10k | gpt-4.1 | 0.00 [0.00, 0.20] | 1.00 [0.80, 1.00] | 4.0 | 2.9 | finished (15/15) |
| latex_math_edit | 10k | grok-4-1-fast-reasoning | 0.93 [0.70, 0.99] | 0.07 [0.01, 0.30] | 5.1 | 31.6 | finished (15/15) |
| latex_math_edit | 20k | gpt-4.1 | 0.00 [0.00, 0.20] | 1.00 [0.80, 1.00] | 4.1 | 3.2 | finished (15/15) |
| latex_math_edit | 20k | grok-4-1-fast-reasoning | 1.00 [0.80, 1.00] | 0.00 [0.00, 0.20] | 5.0 | 37.2 | finished (14/15) |
| latex_xref_update | 10k | gpt-4.1 | 0.13 [0.04, 0.38] | 0.87 [0.62, 0.96] | 6.0 | 5.6 | finished (15/15) |
| latex_xref_update | 10k | grok-4-1-fast-reasoning | 0.53 [0.30, 0.75] | 0.47 [0.25, 0.70] | 6.3 | 70.1 | finished (14/15) |
| latex_xref_update | 20k | gpt-4.1 | 0.00 [0.00, 0.20] | 1.00 [0.80, 1.00] | 6.0 | 5.5 | finished (15/15) |
| latex_xref_update | 20k | grok-4-1-fast-reasoning | 0.60 [0.36, 0.80] | 0.40 [0.20, 0.64] | 6.9 | 88.6 | finished (13/15) |
| para_consolidate | 10k | gpt-4.1 | 0.00 [0.00, 0.20] | 0.67 [0.42, 0.85] | 5.0 | 5.5 | finished (15/15) |
| para_consolidate | 10k | grok-4-1-fast-reasoning | 0.87 [0.62, 0.96] | 0.00 [0.00, 0.20] | 7.6 | 115.6 | finished (13/15) |
| para_consolidate | 20k | gpt-4.1 | 0.00 [0.00, 0.20] | 0.73 [0.48, 0.89] | 5.1 | 5.8 | finished (15/15) |
| para_consolidate | 20k | grok-4-1-fast-reasoning | 0.87 [0.62, 0.96] | 0.00 [0.00, 0.20] | 7.9 | 116.0 | finished (12/15) |
| para_edit | 2k | gpt-4.1 | 0.00 [0.00, 0.20] | 1.00 [0.80, 1.00] | 4.5 | 3.4 | finished (15/15) |
| para_edit | 2k | grok-4-1-fast-reasoning | 0.73 [0.48, 0.89] | 0.27 [0.11, 0.52] | 7.7 | 62.0 | finished (11/15) |

## E.7 Worker Output Guardrails

Some models occasionally violate worker output contracts (for example, wrapper artifacts or invalid paragraph boundaries). We report guardrail ablations here.

Table 37: Worker output guardrails ablation (Functional Safety only; `grok-4-1-fast-reasoning`).

| Task | Size | FS Strict Success (Guardrails Off) | FS Strict Success (Default) | FS Strict Side-Effect (Guardrails Off) | FS Strict Side-Effect (Default) | N |
|---|---|---|---|---|---|---|
| Paragraph edit (sections-only) | 2k | 1.00 [0.80, 1.00] | 0.93 [0.70, 0.99] | 0.00 [0.00, 0.20] | 0.07 [0.01, 0.30] | 15 |
| LaTeX relocate | 10k | 0.93 [0.70, 0.99] | 0.87 [0.62, 0.96] | 0.07 [0.01, 0.30] | 0.13 [0.04, 0.38] | 15 |
| LaTeX relocate | 20k | 0.80 [0.55, 0.93] | 0.93 [0.70, 0.99] | 0.20 [0.07, 0.45] | 0.07 [0.01, 0.30] | 15 |
| Compound ops | 20k | 0.93 [0.70, 0.99] | 1.00 [0.80, 1.00] | 0.00 [0.00, 0.20] | 0.00 [0.00, 0.20] | 15 |
| Compound ops (targeted) | 20k | 0.93 [0.70, 0.99] | 0.80 [0.55, 0.93] | 0.07 [0.01, 0.30] | 0.07 [0.01, 0.30] | 15 |
| Compound ops (sequential) | 20k | 1.00 [0.80, 1.00] | 0.87 [0.62, 0.96] | 0.00 [0.00, 0.20] | 0.13 [0.04, 0.38] | 15 |

Table 38: Contextual Insert guardrails ablation (Functional Safety only; `gpt-4.1`).

| Setting | Success (WS) | Success (Strict) | Side-Effect (WS) | Side-Effect (Strict) | N |
|---|---|---|---|---|---|
| FS (guardrails=off) | 0.93 [0.70, 0.99] | 0.93 [0.70, 0.99] | 0.00 [0.00, 0.20] | 0.00 [0.00, 0.20] | 15 |
| FS (default: retry=1) | 1.00 [0.80, 1.00] | 1.00 [0.80, 1.00] | 0.00 [0.00, 0.20] | 0.00 [0.00, 0.20] | 15 |

# F    Footprint Computation and Payload Preservation Guarantees

**Scope of this appendix.**    Section 4 defines the formal guarantees. This appendix adds only the concrete computation procedures used by the implementation: (1) step-wise structural footprint computation, (2) step-wise payload footprint computation, and (3) a worked multi-step example.

**Operational setup.**    For a hierarchy $R$ and plan $P = (o_1, \ldots, o_k)$, footprints are computed *per operation* on the evolving hierarchy state. We use $F_{\mathrm{struct}}(o_i)$ for nodes whose structural signature changes in step $i$, and $F_{\mathrm{payload}}(o_i)$ for nodes whose local payload may change in step $i$. Plan-level sets are unions across steps:

$$F_{\mathrm{struct}}(P) = \bigcup_i F_{\mathrm{struct}}(o_i), \qquad F_{\mathrm{payload}}(P) = \bigcup_i F_{\mathrm{payload}}(o_i).$$

**Minimal whitespace halo note.**    The payload footprint includes boundary gap nodes and adjacent boundary leaves as a conservative halo so separator canonicalization at edit boundaries is treated as in-scope. This allows expected boundary whitespace movement without weakening non-interference outside the step footprint.

---

**Procedure 4** ComputeStructuralFootprints

---

**Require:** Hierarchy $R$, plan $P = (o_1, \ldots, o_k)$
**Ensure:** Per-op footprints $\{F_{\mathrm{struct}}(o_i)\}$ and union $F_{\mathrm{struct}}(P)$
 1: $H \leftarrow R$; $F_{\mathrm{struct}}(P) \leftarrow \emptyset$
 2: **for** each op $o_i$ in order **do**
 3:     Validate schema and referenced nodes (fail closed on error)
 4:     $H' \leftarrow \mathrm{SIMULATEOP}(H, o_i)$
 5:     $F_{\mathrm{struct}}(o_i) \leftarrow \{n : \mathrm{Sig}_{\mathrm{struct}}(n)$ differs between $H$ and $H'\}$
 6:     $F_{\mathrm{struct}}(P) \leftarrow F_{\mathrm{struct}}(P) \cup F_{\mathrm{struct}}(o_i)$
 7:     $H \leftarrow H'$
 8: **end for**
 9: **return** $\{F_{\mathrm{struct}}(o_i)\}, F_{\mathrm{struct}}(P)$

---

**Procedure 5** ComputePayloadFootprints

---

**Require:** Hierarchy $R$, plan $P = (o_1, \ldots, o_k)$
**Ensure:** Per-op footprints $\{F_{\mathrm{payload}}(o_i)\}$ and union $F_{\mathrm{payload}}(P)$
 1: $H \leftarrow R$; $F_{\mathrm{payload}}(P) \leftarrow \emptyset$
 2: **for** each op $o_i$ in order **do**
 3:     Initialize $F_{\mathrm{payload}}(o_i)$ from edit targets in $o_i$
 4:     Add refreshed subtrees induced by $o_i$ (if any)
 5:     Add moved subtrees, inserted nodes, and deleted nodes
 6:     Add boundary gap nodes and boundary leaves (minimal whitespace halo)
 7:     Add parents/ancestors whose local payload can change due to child-list or gap updates
 8:     $F_{\mathrm{payload}}(P) \leftarrow F_{\mathrm{payload}}(P) \cup F_{\mathrm{payload}}(o_i)$
 9:     $H \leftarrow \mathrm{SIMULATEOP}(H, o_i)$
10: **end for**
11: **return** $\{F_{\mathrm{payload}}(o_i)\}, F_{\mathrm{payload}}(P)$

---

**Worked multi-step example (Move + Insert).**    Move paragraph $p$ from section $A$ to section $B$, then insert a new paragraph under $A$. In step 1, $F_{\mathrm{payload}}(o_1)$ includes the moved subtree, boundary halo at source/destination, and affected parents/ancestors. In step 2, $F_{\mathrm{payload}}(o_2)$ includes the inserted node, boundary halo at the insertion site, and affected parents/ancestors. Payload outside each step footprint remains unchanged for that step; payload outside $F_{\mathrm{payload}}(P)$ remains unchanged after both steps.

**Complexity.** Let $N$ be node count and $k$ be operation count. Worst-case structural scanning is $O(kN)$, with optional $O(N)$ refresh/reconciliation passes depending on execution mode. In practice, edits are usually local, so changed sets are sparse even though the conservative scan is linear.

# G  Formal Methods Checks

## G.1  Bounded model checking of the orchestration loop (TLA+)

In addition to the formal proofs in the main text, we include a small bounded TLA+ model of the hierarchical pipeline's orchestration control loop under nondeterministic worker outputs. The model abstracts a document as a mapping from units to a compact structural token (tracking only a few features relevant to worker-output failure modes, such as wrapper artifacts and top-level heading structure). It models deterministic post-processing (wrapper reapplication and simple header restoration) followed by worker-output guardrails (enabled by default) that can reject malformed outputs and retry once before failing closed.

We model-check a small set of orchestration invariants. For stable cross-referencing, we assign the following invariant IDs (and check them in the TLA+ model):

- **FS-INV-001 (NoSideEffects).** Units outside the explicitly touched set remain byte-for-byte unchanged.

- **FS-INV-002 (NoIntroducedArtifacts).** With worker-output guardrails enabled, an accepted `local_edit` output cannot introduce prompt-wrapper artifacts that were not present in the input (e.g., stray fences, delimiter echoes, or prompt labels).

- **FS-INV-003 (RetryBoundedness).** Worker-output guardrail retries are bounded by configuration; the system fails closed after the last allowed attempt rather than looping.

- **FS-INV-004 (TerminationUnderBounds).** For a finite plan and bounded retries, execution terminates in success or a bounded error state.

We check FS-INV-001 through FS-INV-003 as state invariants, and we check FS-INV-004 as a temporal termination property (under weak fairness).

We check these invariants with the TLC model checker under small, explicit bounds to keep the state space finite while still exercising the guardrail retry and failure semantics. Concretely, we evaluate two bounded configurations: (i) a single-unit case and (ii) a two-unit case. In both, the plan length is bounded to a single operation, guardrails run in `retry` mode with at most one retry per operation, and worker outputs are abstracted to a small finite token space (including a bounded heading-structure feature). TLC finds no counterexample for any of the above properties under these bounds. As with any bounded model checking exercise, these results do not constitute a proof of correctness for unbounded inputs, and they do not model string parsing or the full space of format-dependent extraction behavior; rather, they serve as an additional consistency check on the control-flow logic and guardrail semantics.

**Limitations and out-of-scope realism.** This model is intentionally abstract. In particular, it does not aim to capture several sources of real-world complexity, including: (i) malformed or non-text provider payloads and low-level parsing of raw worker responses (worker outputs are modeled as abstract post-normalization tokens), (ii) format-specific extraction failures and entity-reconciliation details, (iii) planner accuracy or mis-scoping of plans relative to the user's intent (plans are chosen nondeterministically within the configured bounds), and (iv) the full range of operational failure modes (timeouts, rate limits, and other infrastructure errors). We therefore treat the TLC results as a bounded control-flow consistency check, complementary to the proofs and empirical evaluations rather than a complete verification of the deployed system.

