# OpenReview forum: "Functional Safety for Language Models"
_TMLR — Decision pending for TMLR_

### Review · Reviewer_s5mM · 2026-04-21

**Summary Of Contributions:**

This paper introduces Functional Safety (FS), a hierarchy-aware editing architecture that separates LLM-driven editing into a stochastic planning stage and a deterministic execution stage. Documents are represented as explicit hierarchical structures, over which a router emits typed symbolic plans drawn from a five-operator edit algebra. A deterministic executor then applies these plans while enforcing structural invariants, with side-effect scope bounded by formally defined per-step payload and structural footprints. The paper proves a deterministic safety theorem showing that, given a well-formed plan, the executor produces no payload changes outside the declared footprint. Empirically, FS achieves 100% strict success with zero side effects across varied tasks, substantially outperforming ReAct-style tool agents on structural operations.

---

**Key strengths:**

- The formal framework is internally consistent. The footprint definitions, typing rules, and Theorem 2 form a coherent and auditable specification of executor behavior, providing a principled vocabulary for reasoning about side-effect scope in LLM-driven editing.

- The empirical comparison is systematic, and the findings on structural tasks are compelling. The performance gap between FS and ReAct+apply_patch on operations such as section relocation and code visibility refactoring reflects a fundamental limitation of context-matching approaches on flat text, not a metric artifact.

---

**Key weaknesses:**


- Lemma 3 is mathematically vacuous. It bounds side-effect probability by ε without characterizing ε or providing conditions under which it is small, offering no insight beyond a direct application of total probability.

- Hierarchy extraction is a safety-critical dependency that is underemphasized. The formal guarantees break down entirely if extraction fails, yet this is deferred to the limitations section with minimal quantitative treatment.

- The number of distinct documents used across all tasks is very small. Targeted paragraph editing uses only 3 documents per condition, the long-form tasks derive from a single policy brief template, and the LaTeX tasks use a single synthetic book document. This scale is insufficient to draw reliable conclusions about generalization, and the provenance of all prose benchmark documents is never described.

- The model evaluation does not cover sufficiently advanced or recent models. While Appendix B includes local model results for llama3:8b, qwen2.5:14b, and gemma3:27b, more capable recent models such as the Qwen3.5 series are absent entirely, leaving open whether such models already exhibit sufficiently localized editing behavior that reduces the practical advantage of FS.

- The related work discussion is notably sparse with respect to recent literature. The citation list contains few works from 2024-2026. This limits the paper's ability to situate its contributions accurately within the current state of the field.

**Additional Comments:**

N/A

**Audience:**

Yes

**Audience Explanation:**

Yes. Researchers in agentic LLM systems will find the empirical characterization of where ReAct-style agents break down practically valuable. Researchers in formal methods and LLM safety will find the footprint formalism and the planner-executor separation a useful design pattern worth building upon.

**Broader Impact Concerns:**

No significant ethical concerns arise from this work. One point worth noting is that deploying FS in high-stakes settings such as legal drafting or medical documentation while presenting it as providing unconditional formal safety guarantees could create unwarranted confidence among practitioners. The paper should ensure that any user-facing characterization of the system accurately reflects that the guarantees are conditional on planner conformance and correct hierarchy extraction.

**Claims And Evidence:**

No

**Claims Explanation:**

The paper presents both formal and empirical contributions, but there is a gap between the strength of the claims and the evidence provided. On the formal side, the results are technically correct within scope, but Lemma 3 is vacuous (bounding side-effect probability by uncharacterized ε), and hierarchy extraction, a safety-critical dependency, receives minimal quantitative treatment despite extraction failure invalidating all formal guarantees. On the empirical side, the findings are strong on tested benchmarks, but the evaluation scale is insufficient: very few distinct documents, unspecified provenance, only two frontier models tested, local models evaluated only on simple tasks, and sparse engagement with recent literature (2024-2026). This leaves unclear whether results reflect architecture properties or model capabilities and whether advantages persist across the current model spectrum. The authors should either provide substantially more evidence or adjust claims to reflect the limited evaluation scope.

**Requested Changes:**

- Revise or remove the lemma 3. As stated, it seems to contribute no theoretical insight beyond Theorem 2. If retained, ε must be given a concrete characterization.

- Acknowledge hierarchy extraction as a safety-critical dependency in the main text. The current treatment in the limitations section is insufficient given that extraction failure silently invalidates all formal guarantees. Quantitative extraction reliability results on non-templated documents should be included or explicitly scoped out.

- Substantially increase the number and diversity of documents used in the evaluation. Using 3 documents for paragraph editing, a single policy brief template for long-form tasks, and a single LaTeX document is insufficient to support general claims. The provenance of all prose benchmark documents should also be clearly described.

- Include evaluation on more recent and capable models, such as the Qwen3.5 series. As frontier model capabilities improve, it is important to assess whether the advantage of FS diminishes and whether the architecture remains beneficial across the current model capability spectrum.

- Expand the related work section to engage with recent literature from 2024-2026. The current citation list is sparse in this period, and relevant recent work on agentic coding systems, constrained decoding, and structured document understanding should be discussed to accurately situate the paper's contributions.

---

### Review · Reviewer_dAXX · 2026-04-22

**Summary Of Contributions:**

The paper addresses a major concern with LLM editing documents/code. It proposes principled editing methods based dividing the input into hierarchical chunks and then only using the chunks that require editing, thereby preserving everything else that does not need changing. Limitations section is very well-written and it answered a few questions I had while reading the paper. My detailed review:

- Open source results are not in the main paper body, I would like to see them in the same tables as closed-source models, to give readers an idea how the proposed method works with smaller, open models.
- One of the contributions mention 100% success rate while tables 23, 24, 25, etc. show test pass rate <100, this is a bit confusing, please clarify this.
- I would like to see more details on the impact of hierarchy extraction, it seems to be a dependency for good results.
- Detailed ablation studies are lacking, which components from the pipeline matter more and by how much.
- Paper's main body is very thin on citations, I would encourage authors to use reference in the body as needed.
- Is it possible to add more comparators than just ReAct-style agents? any other structure aware approaches? It would be good to at least have this in the discussion as to why not.

**Audience:**

Yes

**Audience Explanation:**

This is an important problem. Especially with the ubiquitous use of LLMs in editing documents and code, having a principled way to only editing what is needed and keeping everything else intact is very useful.

**Broader Impact Concerns:**

I dont have any concerns on the broader impact.

**Claims And Evidence:**

Yes

**Claims Explanation:**

Most claims made in the paper are supported by evidence in the paper.

**Requested Changes:**

- Open source results are not in the main paper body, I would like to see them in the same tables as closed-source models, to give readers an idea how the proposed method works with smaller, open models.
- One of the contributions mention 100% success rate while tables 23, 24, 25, etc. show test pass rate <100, this is a bit confusing, please clarify this.
- I would like to see more details on the impact of hierarchy extraction, it seems to be a dependency for good results.
- Detailed ablation studies are lacking, which components from the pipeline matter more and by how much.
- Paper's main body is very thin on citations, I would encourage authors to use reference in the body as needed.
- Is it possible to add more comparators than just ReAct-style agents? any other structure aware approaches? It would be good to at least have this in the discussion as to why not.

---

### Review · Reviewer_josD · 2026-04-29

**Summary Of Contributions:**

The authors introduce an architecture that formalizes LLM-driven edits as typed plans over explicit hierarchies with
deterministic execution. Finally they validate their findings.

**Additional Comments:**

I think the paper correctly identifies a critical gap in current LLM-based editing systems, i.e., the lack of formal guarantees about edit locality and side effects. The hierarcical representation is appreciable and seems also rigorous. Furthermore, the experiments cover a wide range of tasks and seem consistent. The paper does a good job of bridging  a broader lierature. It seems to me that the fragility of hierarchy extraction in ambiguos texts is underanalysed. MOreover the theoretical guarantees seem somewhat tautological. The main theorem depends on a well-formed plan satisfying footprint containment and structural preconditions, so that it seems that correct scoping is not completely addressed? The paper overall seems an excellent addition to the literature.

In order for this review to be considered fairly I must say that I'm not keeping up on the newest literature on the subject and I am more expert on the mathematical foundations (see regret bounds) for Online Learning. So the confindece in the assesment is not high.

**Audience:**

Yes

**Audience Explanation:**

The topic is of interest for TMLR readers since they takle a problem, bridging also a broader literature of interest, such as works on long-context transformers, hierarchical cognition, tool-augmented language models

**Broader Impact Concerns:**

There are no broader impact concerns.

**Claims And Evidence:**

Yes

**Claims Explanation:**

The authors provide all the material to support their claims.

**Requested Changes:**

There is a minor problem, the  hyperlinks to the theorems are wrong, foe example clickin on theorem 2 you go to lemma 2. See additional comments for the wider opinion on the paper.

---

### Author Response · Authors · 2026-05-12

We are submitting a revised manuscript that reflects the thoughtful comments from the three anonymous reviewers. We are grateful for the time they took to read the paper carefully. Their feedback helped us sharpen the claims, improve the presentation of the empirical results, and make the limitations more explicit.

The main revision is that the claims are now less absolute. We emphasize that Functional Safety provides conditional guarantees: when the planner emits a valid plan, and when local worker outputs satisfy the required interface contracts, the executor can enforce locality properties. The architecture can still fail through planner or worker errors. Some failures are detectable, such as malformed JSON or rejected worker outputs, but others remain possible, including cases where the planner selects the wrong edit scope. We now discuss these failure modes more directly.

We also revised the experiments section. The metrics and terminology have been tightened. Single-document success is defined as task completion with no forbidden side effects under the reported metric. Multi-file test pass is treated separately as a downstream unit-test metric, not as the same quantity as strict single-document success. Throughout the paper, broad claims have been replaced with claims about observed success and observed side-effect rates under the evaluated conditions.

In response to Reviewer 1, we removed the probabilistic lemma, which did not add enough value. The formal section now focuses on operator-local payload locality and a deterministic safety theorem for validated plans and accepted local worker outputs. This gives a cleaner statement regarding safety guarantees.

We also expanded the discussion of hierarchy extraction. The revised paper treats extraction as a safety-critical dependency rather than a minor implementation detail. It distinguishes parser-backed structured documents from weakly structured or ambiguous inputs, and it adds a small weakly structured prose pilot using deterministic paragraph anchoring. We present that pilot as evidence that conservative anchoring can be useful, not as evidence that arbitrary unstructured extraction is solved.

To improve empirical confidence, we expanded the remote API-based evaluation and added Qwen3.5 to the local-model results. The local-model results are broadly consistent with the other evaluated local models of similar size.

We also attempted to add DeepSeek results. However, in our runs the DeepSeek API sometimes continued processing for an extended period and then returned no usable result, without an error code or clear timeout indication. Because we could not obtain a complete and reliable comparative run under the same evaluation protocol, we excluded DeepSeek from the main paper rather than treating incomplete API traces as evidence.

---

### Author Response · Authors · 2026-05-13
**Sample size**

We agree with the anonymous reviewer who pointed out that repeated model calls on a small number of document instances can induce within-document correlation, so the effective evidence is not captured by the raw run count alone. In response, we expanded the single-document evaluation to use 10 document IDs per condition with five repetitions per document. The aggregate sample size varies by task family because some families contain more task/size/model conditions, but every headline condition now uses the same 10-document, five-repetition design. The expanded results are close to the original estimates and do not change the paper’s main conclusions, but they reduce the concern that the observed patterns are artifacts of a small set of documents.